# What are the prospects for seasonal prediction of the marine environment of the Northwest European shelf?

**Abstract.**

Sustainable management and utilisation of the Northwest European Shelf seas (NWS) could benefit from reliable forecasts of the marine environment on monthly-to-seasonal timescales. Recent advances in global seasonal forecast systems, and regional marine reanalyses for the NWS, allow us to investigate the potential for seasonal forecasts of the state of the NWS. We identify three possible approaches to address this issue: A) basing NWS seasonal forecasts directly on output from the Met Office's GloSea5 global seasonal forecast system; B) developing empirical downscaling relationships between large-scale climate drivers predicted by GloSea5, and the state of the NWS; and C) dynamically downscaling GloSea5 using a regional model. We show that the GloSea5 system can be inadequate for simulating the NWS directly and so move on from A). Turning to B), we explore empirical relationships between the winter North Atlantic Oscillation (NAO), and NWS variables estimated using a regional reanalysis. We find some statistically significant relationships, and present a skilful prototype seasonal forecast for English Channel sea surface temperature.

We then consider the potential of C). We find large-scale relationships between inter-annual variability in the boundary conditions and inter-annual variability modelled on the shelf, suggesting that dynamic downscaling may be possible. We also show that for some variables there are opposing mechanisms correlated to the NAO, for which dynamic downscaling may improve on the skill possible with empirical forecasts. We conclude that there is potential for the development of reliable seasonal forecasts for the NWS, and consider the research priorities for their development.

## 1. Introduction

### 1.1. Background

The Northwest European Shelf seas (NWS) are of wide economic, environmental and political importance. They support many ecosystem services and human activities, including fisheries, energy extraction and transmission (both renewable and non-renewable), shipping and waste removal. Most of these services and activities are sensitive to the variable environmental conditions under which they operate, for example:

- Shipping (transport and industrial) and offshore oil/gas and renewable operations are sensitive to wind/wave conditions and currents;
- The capacity of sea-floor gas distribution networks is sensitive to bottom temperature (with capacity decreased in cold conditions when demand is likely to be highest;
- Commercial and recreational fisheries are sensitive to large-scale primary production and the seasonal evolution of the marine food web, which in turn are sensitive to surface temperature, salinity and seasonal re-stratification (see e.g. Fernandes, 2015);
- Some specific commercial species have a life cycle or food web which is sensitive to near-bottom temperature (see e.g. Dulvy et al., 2008; Jones et al., 2015; Pinnegar et al., 2017; Pörtner and Farrell, 2008; Pörtner and Peck, 2010);

- Coastal installation operations may be sensitive to local flooding (surge events), to sea temperatures (e.g. ambient temperatures for power station cooling), and to consequential local ecosystem impacts (Brown et al., 2016; Dawson et al., 2016; Schneider et al., 2015).

Because of these sensitivities, the ability to predict variations in the marine environment is of great potential value for marine operations, management, planning and conservation. Weather and wave forecasting are of course well-established tools, and more recently operational forecasting of marine environmental variables such as temperature, salinity and currents for a few days ahead has matured to the extent that daily forecasts are now widely and freely available (e.g. for European regional seas through the Copernicus Marine Environment Monitoring Service http://marine.copernicus.eu/). Centennial climate projections are also available for the NWS (e.g. Tinker et al., 2016), but there is an important gap between these timescales. Over the last 10 years significant progress has been made on prediction at multi-annual to decadal time-scales (Hughes et al., 2017; McCarthy et al., 2017; MCCIP, 2017), that may be particularly compatible with policy and legislation review and reporting cycles (Frost et al., 2016) as well as business planning and investment timeframes. In this paper we examine the prospects for prediction of the marine environment on a timeframe of 1-6 months ahead.

One can envisage many potential applications for marine predictions at this extended range. For example, setting and management of fishing quotas for maximum sustainable yield could take account of likely conditions for recruitment in that year, energy producers and suppliers could anticipate winter seasons with higher than usual demand and/or stress on the offshore supply network, and environmental regulators could use early warning of regions at increased risk of water quality issues to target monitoring resources at those regions. But first we must demonstrate that a useful level of predictive skill is achievable on these timescales. Of course, on the 3-6-month timescale, environmental variations are dominated by the seasonal cycle, and this is normally factored into existing decision-making processes. Here we are asking whether we can predict anomalies relative to the average seasonal cycle (e.g. unusually cold winters or unusually stormy autumns).

### 1.2. State of the art and possible approaches to NWS seasonal forecasts

Recent progress in climate modelling and prediction has shown that skilful predictions are possible several months ahead for some key elements of European climate. For example, Scaife *et al.,* (2014) show substantial skill in predicting the winter North Atlantic Oscillation (NAO) index 1-4 months ahead. There is even significant skill in predicting the winter NAO index one year ahead (Dunstone et al., 2016). The NAO is a key determinant of the nature of the European winter (e.g. Hurrell, 1995) with a positive NAO index indicating mild, wet conditions and a negative NAO index indicating cold, dry conditions. This predictive skill has been exploited to demonstrate predictability in a number of user-relevant variables, e.g. relating the phase of the NAO index to the likely number of transport disruption (road, rail and air) impacts (Palin et al., 2016), but so far there has been less attention to seasonal predictability of marine variables (e.g. Hobday et al., 2016) .

To address the problem of seasonal prediction for the NWS we must consider how to derive marine variables from the seasonal climate predictions. The seasonal predictions are produced from global ocean-atmosphere climate models which are initialised with observed conditions and run forward in time through the 1-6 month forecast period. Because the climate is not perfectly predictable on these timescales an ensemble of runs is used to derive a probabilistic prediction. To turn this into a prediction for NWS marine variables, three possible approaches can be envisaged:

A. Read off the NWS marine variables (temperature, salinity, currents) directly from the underlying climate model;
B. Use observations to derive empirical relationships between large-scale climate indices (e.g. NAO index) and the marine quantity of interest, for instance this could be for a fishery metric like Cod recruitment (Engelhard et al., 2014; Stige et al., 2006) or Squid abundance (van der Kooij et al., 2016), then input the forecast climate index to the empirical model to get a forecast of the quantity of interest ('empirical downscaling');
C. Use the marine and atmospheric variables predicted by the climate model as inputs to a higher resolution regional model of the NWS ('dynamical downscaling').

Approach (A) is appealingly simple but is unlikely to be successful. The ocean resolution of the global seasonal forecast system GloSea5 used by Scaife *et al.,* (2014) is around 20 km in the NWS region, with 18 vertical levels in the top 50 m. While this is high by the standards of current global seasonal prediction systems it is still insufficient to resolve many features of the NWS circulation. Further, in common with most global climate models the GloSea5 system does not represent key shelf seas processes such as tidal mixing. While this has been shown to be a successful approach in other regions (e.g. Hobday et al., 2016 used this approach in Australia) we will show later that this approach (A) can be inapplicable to the NWS under some conditions.

### 1.3. Scope of this study

A number of scientific and technical developments have recently come together to enable us, for the first time, to assess the potential for seasonal forecasts for the NWS: first, the demonstration of skill of global seasonal forecasting systems in predicting key European climate indices several months ahead (e.g. Scaife et al., 2014); secondly the development of regional oceanographic models of the NWS, proven for use in operational prediction (O'Dea et al., 2012); and thirdly the combination of the regional NWS models with historical observations, to produce a consistent estimate of the time-varying state of the NWS over recent decades ('regional NWS reanalysis', Wakelin et al., 2014). The NWS regional model allows us to investigate the prerequisites of the dynamical downscaling approach (C), while the reanalysis allows us to investigate in detail the mechanisms of year-to-year variability, and so evaluate which elements of the NWS state are likely to be predictable. For the mean climate, the dynamical downscaling approach (C) has been shown to add value to the simple approach (A) of reading off variables from the underlying global climate model (e.g. Mathis et al., 2013).

In this study we use the above building blocks to evaluate the potential of seasonal predictability for the NWS. We address the following questions in turn:

1) How well does the global seasonal prediction system GloSea5 represent the state and inter-annual variability of the NWS? (Approach A);

2) Can a regional reanalysis adequately represent the inter-annual variability of the NWS? If so can we use it as a proxy to understand mechanisms of variability in the real world?

3) Do the predictable climate indices (e.g. NAO index) provide actual predictive skill for the NWS? (Approach B);

4) What are the prospects for improving NWS seasonal forecasts? (including prerequisites for Approach C).

From the answers to (1-4), we conclude by assessing the prospects for seasonal prediction on the NWS and the likely pros and cons of the direct approach (A), and empirical and dynamical downscaling approaches (B) and (C), and suggest near-term priorities for applications and research.

Our aim in this paper is to draw evidence on the prospects of seasonal forecasting from the literature and from our own research. We recognise that some aspects will not come as a surprise to some readers, nonetheless, we believe that an overall perspective hasn't been presented before.

### 2. Method

We use data from two modelling systems (NWS reanalysis (CO5) and GloSea5) and several observation datasets. Firstly, we describe and compare the two modelling systems, before introducing the observations datasets and the analysis techniques.

### 2.1. NWS Reanalysis (CO5)

The Met Office provides a reanalysis of the NW European shelf seas to the Copernicus Marine Environment Monitoring Service: this has been extensively described and validated (O'Dea et al., 2012; Wakelin et al., 2014), but here we give a brief overview.

The NWS Reanalysis is based on the NEMO Coastal Ocean model version 5 (CO5) implementation (Figure 1). This is on a regional 7 km grid extending from 40°4' N, 19° W to 65° N 13° E, with 50 terrain following levels (s-levels, Siddorn and

Furner, 2013). The simulations run from 1983-2013. The model surface forcings were calculated with the Coordinated Ocean Research Experiments (CORE) bulk formulae (Large and Yeager, 2009), using ERA Interim data (ERAI; Dee et al., 2011). The ocean lateral boundary forcings before 1990 were taken from a simulation from the Forecasting Ocean Assimilation Model (FOAM, Bell et al., 2000), after which they were taken from the global reanalysis used to initialise the

5 GloSea5 seasonal forecasting system. The river forcings were taken from the E-HYPE dataset (Donnelly et al., 2013) and include inter-annual variations. The CO5 reanalysis assimilates Sea Surface Temperature (SST) from satellites.

The use of CORE bulk formulae will have a few important implications for this study. For example, the bulk formulae assume that the surface air temperature has an infinite heat reservoir, and so the SST will tend to follow the surface air temperature, rather than the opposite which occurs in reality. This will affect the relationship between the SST and surface

air temperature. Furthermore, the CORE bulk formulae receive prescribed downward component of the short-wave- and long-wave- radiation, and calculate the upward component internally using modelled SST. Therefore, while SST data assimilation will improve the SST, and affect the upward radiation, the downward radiation is not affected by the SST assimilation. In sections 3.3 and 3.4 of this study, we consider correlations of shelf sea variables with the prescribed *downward* component of the radiative fluxes only.

**2.2. GloSea5**

The GloSea5 seasonal forecast system is described in detail by MacLachlan *et al.* (2014). Here we give a brief description, and focus on the difference between its ocean component and that of the CO5 NWS Reanalysis. GloSea5 is based on the Met Office Hadley Centre climate model HadGEM3-GC2 (Williams et al., 2015). This is a coupled climate model combining the MetUM atmosphere model (N216, ~0.7° horizontal resolution, Brown et al., 2012; Walters et al., 2011), the ocean model

NEMO (Megann et al., 2014), the land surface scheme JULES (Best et al., 2011), and the sea-ice model CICE (Hunke and Lipscomb, 2010). The ocean model component is run on the ORCA025 grid – a 0.25° tri-polar grid (~27 km at the equator), with 75 horizontal z layers (of which 18 (24) are within the top 50 m (100 m)). NEMO ORCA025 is run with a data analysis system (3D-Var) to assimilate a range of observations, including SST (in-situ and along track satellite, mostly AVHRR (Pathfinder) and (A)ATSR (ESA), although some AMSRE are used during the GHRSST period), sea-surface height

(altimetry from AVISOv3 along track), sea ice concentration (OSI-SAF), and water column structure (ARGO floats).

To make seasonal forecasts, a set of GloSea5 simulations are run to form a forecast ensemble and re-forecast (hindcast) ensemble. Every day, 2 ensemble members are initialised and run forward for 216 days. The previous 3-weeks are combined into a 42-member ensemble to make a 6-month forecast, which is updated weekly. A re-forecast ensemble is also run every week (with the same modelling system) to correct bias and drift in the forecasts. This includes 4 start dates for the relevant

30 month, for each of the previous 23 years, run forward for 216 days. The forecast ensemble is used to predict how the following 6 months will compare to this climatology. The system is also run as a continuous reanalysis (the GloSea5 ocean and sea-ice global reanalysis) from 1990-2015 to provide initial conditions for the ocean component of the hindcast ensemble. The atmospheric initial conditions for the forecast are taken from the Met Office operational weather forecast system, and the hindcast atmosphere is initialised from ERA-interim.

GloSea5 shows improved year-to-year predictions of the major modes of variability compared to the previous system (GloSea4, Arribas et al., 2011). Predictions of the El Niño–Southern Oscillation are improved with reduced errors in the West Pacific. GloSea5 shows unprecedented levels of forecast skill and reliability for both the North Atlantic Oscillation and the Arctic Oscillation (MacLachlan et al., 2014).

**2.3. Comparison between NWS Reanalysis and GloSea5**

The GloSea5 ocean and sea-ice global reanalysis and CO5 NWS reanalysis differ in a number of ways. Both rely on the ocean model NEMO, but run with grids of different horizontal and vertical resolutions. The GloSea5 global reanalysis

system is a coupled global model system designed to capture the key components of the global climate system in order to make a seasonal forecast, having used data assimilation over a wide range of variables to constrain the model. Conversely, the CO5 NWS reanalysis is a regional reanalysis, where a higher resolution ocean model is (one-way) forced from ERA Interim atmosphere forcing, using SST-only data assimilation. Effectively, the CO5 NWS reanalysis has a higher resolution and better representation of the NWS physics, whereas the GloSea5 global reanalysis has global scope and assimilates a wider range of observations. Both have high enough resolution to include an open Dover Strait allowing a route for Atlantic water into the southern North Sea which is important for simulating the local seasonal cycle of salinity.

The CO5 NWS reanalysis requires lateral boundary conditions (GloSea5 is a global model system and so doesn't need them) which are taken from FOAM before 1990 and from the GloSea5 Reanalysis thereafter. The change from FOAM to the GloSea5 Reanalysis leads to a discontinuity in the lateral boundary conditions that is important for variables such as sub-surface temperature and salinity in the open ocean. While neither sub-surface temperature nor salinity are considered in this study, they may influence NWS properties (such as SSS and SST) that are considered.

An important difference between the two systems is that the CO5 NWS reanalysis is a shelf seas model that includes all the key shelf seas processes whereas the ORCA025, being a global model, neglects some regionally important processes, including dynamic tides. Tides are particularly important in this region, as the NWS contributes significantly to the global total tidal energy dissipation (Egbert and Ray, 2001). While tides are modelled directly within the CO5 NWS reanalysis, tidal mixing is parameterised in GloSea5. This parameterisation (Simmons et al., 2004) is based on a climatology of turbulence associated with internal tide wave breaking and steep bathymetry (which is very low on the NWS), thus tidal mixing is much less than in the CO5 NWS reanalysis.

There are also key differences in the riverine forcings, and the treatment of the Baltic Sea. The CO5 NWS reanalysis uses river forcings (with inter-annual variability) from the E-HYPE river model (which gives too much discharge), and treat the Baltic as an open boundary where T and S are relaxed to output from a Swedish Meteorological and Hydrological Institute (SMHI) model. GloSea5 uses a river climatology, and models the Baltic explicitly (although at too coarse a resolution to accurately simulate the complex interaction between the Baltic and NWS shelf sea). Both the river and Baltic climatologies will dampen the interannual variability of salinity, particularly near major river outflows, and downstream of the Baltic (in the Skagerrak and in the Norwegian Trench), which will weaken the interannual variability of the associated regional means, and tend to reduce their interannual correlations. The equivalent effect for temperature from the Baltic will be countered by the SST data assimilation.

## 2.4. Observations

Much of the evaluation of the CO5 NWS reanalysis (Wakelin et al., 2014) focused on the mean state of the model. Here we are more interested in the modelled temporal variability, and so undertake additional evaluation. We compare the model to limited observed time-series to assess its performance at replicating several observed events. Here we describe the observed time-series.

### 2.4.1. Southern North Sea Ferry data

Ferries are well established vessels of opportunity for oceanographic measurements taking regular long-term samples of surface water while the ship is on passage between ports (Bean et al., 2017). Observations can be in the form of samples taken by crew for subsequent testing in a laboratory or more sophisticated "Ferry boxes" as packages of instruments that semi-autonomously monitor temperature, salinity and other water properties. We use the monthly salinity data from the ferry on the Harwich to Hook of Holland route, which took quasi-weekly temperature and salinity samples at 9 standard stations between 1971 and 2012 (Joyce, 2006) and is reported in the ICES Report on Ocean Climate (Larsen et al., 2016) and MCCIP Report Cards (Dye et al., 2013). We use point time-series from this dataset to compare to the model.

### 2.4.2. Western Channel Observatory (WCO)

The Western Channel Observatory (WCO) is an oceanographic time-series in the Western English Channel (Smyth et al., 2015). *In situ* measurements are undertaken fortnightly at open shelf station E1 (50.03˚N, 4.37˚W) using the research vessels of the Plymouth Marine Laboratory and the Marine Biological Association. We compare time-series of temperature and salinity from a range of observed depths to model output from the nearest grid box.

### 2.4.3. ICES Report on Ocean Climate data

We use annual-mean time-series data from three stations used in the ICES Report on Ocean Climate (González-Pola et al., 2018): Malin Head weather station; Fair Isle; and Helgoland Roads. Malin Head SST (55.37°N 7.34°W) is provided by the Irish Marine Institute/Met Éireann (Cannaby and Hüsrevoğlu, 2009; Nolan et al., 2010). The Fair Isle time-series (59°N 2°W) is provided by Marine Scotland Science to measure the temperature and salinity (upper 100m) of Atlantic water entering the North Sea via the Fair Isle Current (Hughes et al., 2018). The Helgoland Roads (54.1833°N 7.9°E) time-series is provide by the Alfred Weneger Institut/Helmholtz-Zentrum für Polar und Meeresforschung, and comprises of surface temperature and salinity (Raabe and Wiltshire, 2009; Wiltshire et al., 2015; Wiltshire and Manly, 2004). The data are freely available to download from ICES (https://ocean.ices.dk/iroc/).

### 2.4.4. NAO

The North Atlantic Oscillation (NAO) is a climatic phenomenon in the North Atlantic Ocean of fluctuations in the difference of atmospheric pressure at sea level between the Icelandic low and the Azores high. These fluctuations control the strength and direction of westerly winds and storm tracks across Europe (Hurrell, 1995).

We use the NOAA National Weather Service Climate Prediction Center NAO data (http://www.cpc.noaa.gov/products/precip/CWlink/pna/nao.shtml), for monthly mean NAO index. We only use winter (DJF) for the years 1992/1993 – 2010/2011 to be consistent with the available GloSea5 ocean and sea-ice reanalysis NAO index time-series.

### 2.4.5. Storm track latitude index

When analysing the relationships of the shelf salinity we find correlation patterns which suggest storm track latitude may be important. We therefore analysed the mean sea level pressure data to produce a Storm track latitude index, following a method adapted from Lowe *et al.* (2009). The 3-hourly ERA Interim mean sea level pressure data from all modelled latitudes at 2°30'E were filtered with a Blackman band pass filter. The temporal variance of this filtered mean sea-level pressure was calculated for each month for each grid box at 2°30'E, and the latitude with the greatest variance was recorded as the storm track latitude. We consider the winter (DJF) mean storm track.

### 2.5. Analysis techniques

We use regional mean time-series of model output (sea-surface temperature and salinity (SST, SSS), near-bed temperature and salinity (NBT, NBS), and the difference between surface and near-bed for temperature and salinity (DFT, DFS)) from the reanalysis, adapted from the region mask from Wakelin *et al.* (2012) (Figure 1a). We calculated monthly, seasonal and annual means from these time-series. In addition to model output, we extract regional mean time-series for the ERAI surface forcings, the GloSea5 ocean lateral boundary conditions (using the masks in Figure 1b-d), and the E-HYPE river forcings.

### 2.5.1. Identifying relationships between the NWS response and the drivers.

Important relationships between the shelf drivers and the shelf response are identified by comparing time-series. Noting that a statistically significant correlation does not imply a causal relationship, the spatial patterns of the correlation coefficients

are used to help interpret the underlying mechanisms behind the correlations. Some possible mechanisms are described, but it is considered beyond the scope of this study to undertake sensitivity studies to explore any mechanisms in detail.

The region mask of Wakelin *et al.* (2012) (Figure 1a) is used to create regional mean time-series of the results from the NWS Reanalysis, including its atmospheric (downward radiative fluxes, surface air temperature and relative humidity, total precipitation, mean sea level pressure, wind magnitude) and riverine forcings. The oceanic T and S forcings from around the boundary, are averaged into 21 regions based on horizontal and vertical gradients to T and S (typically dividing the north, west and south-western boundaries into surface, mid-depth and deep layers (according to the typical modelled summer and winter mixed layer depths), and then dividing the boundaries horizontally according to features within the data). A deep layer of salty, relatively warm water in the south-western and the southern part of the western boundary is identified as Mediterranean Intermediate Water and is treated separately. Most correlations have been found with the surface (0 - 30 m) regions, and so this study focuses on these regions. These regions are shown in Figure 1b-d. We also use annual and monthly mean time-series of the NAO and Storm Track latitude.

Model and observed time-series are compared to one another with Pearson's correlations, and their significance is noted at the 95 % confidence level. Typically, we compare the annual mean time-series, but we also compare at the seasonal time-scale. We also investigate lag-correlations between the shelf response, and possible drivers and climate indices. For example, the time-series of DJF NAO will be correlated with the DJF SST across the shelf (at 0-months lag). The DJF will then be compared to the JFM SST (January-March; 1-month lag), FMA (February-April; 2-month lag) etc. (e.g. Figure 6).

For consistency we have used the same region mask (e.g. Wakelin et al., 2012) for the river forcings as for the shelf seas variables and surface atmospheric forcings. However, this mask was not designed for rivers and so several regions must be treated with care, or excluded. For example, the northern North Sea region combines the river flow from small sections of the Scottish and Danish coasts which does not make sense – other regions to be excluded are the central North Sea, Shetland shelf region, and the North Atlantic regions. Other regions combine river flow from different coasts, but in a more sensible manner – for example the English Channel and Irish Sea regions combine river input from two coasts, but due to the smaller enclosed nature of these regions, this is sensible in terms of local salinity. In the modelling system, the rivers do not have a specified temperature, and so assume the local temperature when they reach the sea. Therefore, rivers predominantly affect salinity, with only secondary temperature effects (associated with changes in density driven circulation and stability). As increased river flow reduces the local salinity, most correlations are expected to be negative.

## 3. Results

### 3.1. Question 1: How well does the GloSea5 global seasonal prediction system represent the NWS?

Both GloSea5 and the CO5 NWS reanalysis assimilate similar SST observations, and so their simulated patterns of SST are relatively similar, and in agreement with observations. When looking at the near-bed temperatures (NBT) and the difference between the surface and bed temperatures (DFT), there are important differences between GloSea5 and the NWS Reanalysis. DFT is an important diagnostic of stratification – when DFT > 0.5 °C the water column is considered stratified, and the DFT = 0.5 °C isotherm is indicative of the location of the modelled tidal mixing fronts.

A time-series showing the seasonal cycle of areal extent of the stratification (from a 20-year mean) for both GloSea5 and the CO5 NWS reanalysis is presented in Figure 2a. The CO5 NWS reanalysis has been shown to have a generally good representation of the seasonal stratification where independent observations are available (O'Dea et al., 2012). While both the CO5 NWS reanalysis and GloSea5 show that the NWS is fully mixed in the winter (effectively no grid boxes are stratified), in GloSea5 more grid boxes are stratified in the summer than in the NWS Reanalysis: from April to September, the stratified area of the shelf is ~20% more in GloSea5 compared to NWS Reanalysis. Figure 2b shows a map of the stratified regions for May (an exemplar stratified month). This shows that much of the southern North Sea, English Channel and Irish Sea that is modelled as being mixed in May in the NWS Reanalysis, is stratified in GloSea5, due to insufficient

turbulence within the GloSea5 NWS (lack of tidal mixing) common to most global ocean models. This highlights an important weakness in using the GloSea5 system to provide direct information on the NWS.

Further evidence is shown under Question 2 below that using GloSea5 NWS fields directly can be problematic. We therefore conclude that this approach (Approach A) may not be appropriate under some conditions, in which case, some form of

(dynamic or empirical) downscaling of the GloSea5 fields may be needed to generate reliable NWS forecasts.

### 3.2.  Question 2: How well does the CO5 NWS reanalysis represent inter-annual variability on the NWS?

While the ability of the CO5 NWS reanalysis to simulate the mean state of the NWS is thoroughly evaluated (O'Dea et al., 2012; Wakelin et al., 2014), its ability to simulate inter-annual variability has received less attention. Evaluation requires long observed time-series, preferably of variables that are not assimilated into the reanalysis. Here we focus on 5 locations

across the NWS: the Harwich to Hook of Holland ferry box in the southern North Sea; the Western Channel Observatory (WCO) in the English Channel; the Malin Head weather station, north of Ireland; the Fair Isle time-series (northeast of Scotland), and then Helgoland Roads time-series in the southern North Sea. All these datasets are described in the Methods section.

First, we compare the observed time-series of surface salinity in the southern North Sea (from the Ferry Box) to that from

GloSea5 and the CO5 NWS reanalysis (Figure 3, both from nearest model grid-box). The time-series exhibits multi-year oscillations and these are well simulated by the NWS Reanalysis (r = 0.89, p = 0.00), despite the fact that it does not assimilate salinity observations. There is a fresh bias in the model (-0.20 psu) and a slightly greater variation (standard deviation ratio of 1.11). GloSea5 does not capture a realistic multi-annual variability (r = 0.18, p = 0.62 with standard deviation ratio of 1.81) and modelled salinity also shows a large fresh bias that increases due to a substantial salinity drift

over the duration of the time-series – further evidence that direct reading from NWS fields from GloSea5 would be problematic. As there are differences in the river forcings between GloSea5 and the CO5 NWS reanalysis we would expect differences in the modelled salinity. The CO5 NWS reanalysis uses E-HYPE river forcing (Donnelly et al., 2013) which are specified daily whereas GloSea5 uses a river climatology (Bourdalle-Badie and Treguier, 2006; Dai and Trenberth, 2002), and so exhibits no inter-annual variability.

Secondly, we compare the observed WCO temperature and salinity profiles to the daily mean of the nearest CO5 NWS reanalysis grid box (Figure 4). The WCO observations are not assimilated into the NWS Reanalysis, but the SST from complementary satellite products is. Unsurprisingly, the CO5 NWS reanalysis SST is in close agreement with the WCO observations, for both the seasonal cycle, and the year to year variations at the surface and at depth (30 m). The inter-annual variability is well captured in SST for all seasons (r > 0.99, p = 0.000), and for most seasons at 30 m (typically r > 0.9, but

September – November r = 0.59). There is little seasonal cycle, trend or inter-annual variability in the WCO salinity (e.g. compare the inter-annual variability in Figure 3 and Figure 4). Given the lack of salinity seasonality, we compare all the WCO-reanalysis data pairs, which has a significant correlation of r = 0.49 at the surface and r = 0.65 at 30 m (p = 0.000 for both cases).

The ICES data set includes annual mean data from three very different sites (Figure 5). As we are most interested in the

modelled variability, we evaluate against the anomalies (time-series minus mean) and consider the standard deviations and correlations.

The Malin Head time-series show a good agreement between the reanalysis and observed SST, with a significant Pearson's correlation of r = 0.87, and a relative standard deviation of rsd = 1.13 (model standard deviation (0.47 °C) divided by the observations standard deviation (0.42°C)). These results are remarkably good, but may simply reflect that the model

assimilates SST.

The Fair Isle time-series monitors the properties of one of the main pathways of Atlantic water into the North Sea, so evaluation is important to lend credibility to the results of this study. Here the observations are for the upper 100m, and so

the data assimilation is less dominant in this comparison. As with the Malin Head data, there is a very good agreement with the observations, with a significant correlation of r = 0.86, and relative standard deviation of 1.15, and a bias of 0.75°C (model too warm). The salinity evaluation is also good, with a significant correlation of r = 0.40, a relative standard deviation of 1.01, and a bias of 0.46 psu. Given the timing and frequency of many of the peaks and troughs, there appears to be a better (visual) agreement than is reflected by this correlations value.

The Helgoland Roads time-series shows an excellent temperature agreement (r = 0.99, rsd = 0.98, bias = -0.09°C). The salinity time-series' significant correlation with the reanalysis salinity (r = 0.69) and relative standard deviation of 0.88, suggests a very good representation of the variability. There is a large salinity bias (2.26 psu) however, reflecting the large spatial gradients within this region, and perhaps less riverine influence in the reanalysis.

There are two other ICES time-series in the vicinity of the North Sea, Utsira B, within the Norwegian Trench, and the Norwegian site Svinoy, off the shelf to the north of the Norwegian Trench, both in regions which are known to be difficult to model, and outside the main focus of the Reanalysis. Both showed much lower modelled salinity variability, and low correlations.

Further evaluation of the CO5 NWS reanalysis against other long time-series is planned through the Copernicus NWS regional Marine Forecasting Centre. These initial results suggest that the reanalysis can provide valuable information on inter-annual variability on the NWS, where there is a strong signal.

We now investigate empirical forecasts based on the response of the CMEMS reanalysis to the observed NAO, and then applied to the GloSea5 forecast NAO.

### 3.3.  Question 3: Can predictable climate indices provide real predictive skill for the NWS?

In the literature, there are many empirical relationships between climate indices and various physical and biological responses. The CMEMS reanalysis (through data assimilation) combines observations with models to give the best possible state estimate of the NWS, and so provides a powerful tool to develop such relationships. We focus on the winter NAO, as it is an important source of year-to-year variability in the NWS, and GloSea5 has predictive skill for it. By investigating relationships between the CMEMS reanalysis fields and the observed (NOAA) NAO, and then considering how these relationships change when we use the GloSea5 forecast NAO, we can explore the empirical approach to NWS seasonal forecasting. We note that many of the relationship we find between the NAO and the NWS are not new, and are underlain by published relationships (e.g. Becker and Pauly, 1996; Dippner, 1997; Hurrell and Deser, 2009; Hurrell and Van Loon, 1997). First we focus on shelf temperature, and restrict our analysis to the surface forcing that we consider important for shelf temperatures. We investigate the correlations (and lagged correlations) between the winter (DJF) NAO and this subset of surface forcing (Figure 6). We find a positive correlation of the NAO with the DJF surface air temperature (consistent with Hurrell and Van Loon, 1997) (Figure 6a) and humidity, (not shown), and this persists for one month (to January-March) in most regions (SAT in Figure 6b). This reflects one of the main characteristics of a NAO positive winter – warmer, wetter winters over northern Europe, with cooler, drier winter over southern Europe. There is a significant negative correlation between DJF NAO and the incoming solar radiation for shelf regions west of the UK (SSRD in Figure 6g), although this only persists to a significant level in the Irish Shelf (Figure 6h). DJF NAO is strongly positively correlated to the downward component of thermal (long-wave) radiation for most shelf regions west of the UK and for the Norwegian Trench (NT), and this persists for 3 months in some regions (English Channel) (STRD in Figure 6d-f). Under NAO negative winters, there are more cold, clear days with greater downward solar radiation, compared to NAO positive years, hence the negative correlations (between NAO and downward solar radiation). During positive NAO winters, the greater cloud cover reduces the downward solar radiation, and increases the downward thermal radiation from clouds and water vapour – this supporting the positive correlations between the NAO and downward thermal radiation. Note that the SST assimilation increments will affect the upward radiation fluxes (which are calculated from the modelled SST), but, as the reanalysis system is an ocean-

only uncoupled system, the downward component of the radiation (which we consider here) is prescribed (by the ERA interim data), and so these correlations are not affected by this assimilation.

We now consider the surface forcings we think are likely to be important for shelf salinity (Figure 7). The DJF NAO is strongly correlated with the DJF 10 m wind magnitude (UV10; defined as the magnitude (wind-speed irrespective of direction) of the 10m wind) across the domain (consistent with Hurrell and Deser, 2009), and this persists into the third month (March-May) for the southern and central North Sea (UV10 in Figure 7d-f). The correlation between winter (DJF) NAO acts in opposite ways for winter (DJF) mean sea level pressure (MSLP) and for total precipitation (not shown). The DJF mean sea-level pressure (total precipitation) is negatively (positively) correlated with DJF NAO in the northern regions and positively (negatively) correlated in the southern regions. These correlations persist for a few months in some regions (Figure 7a-c), and are consistent with the correlations of Hurrell and Deser (2009). The observed north-south gradient in correlations between MSLP and NAO reflects the pressure gradient nature of the NAO. The stronger winds associated with the NAO positive phase lead to the positive correlations with wind across the domain. The north-south dipole in wetter/drier weather (between northern and southern Europe) lead to the north-south gradient in correlation between NAO and total precipitation.

River systems can give additional predictability by continuing to respond after the forcing, or can reduce predictability by having such long response times that they act as a low-pass filter. Furthermore, different river catchment areas are located in different climate regimes, and so can respond in different ways, which can further complicate the response. The river runoff forcings in the Norwegian Trench (Figure 7g-i) are highly correlated with the DJF NAO, and this persists until the following summer (July-September, not shown). The runoff in regions influenced by northern British and Irish rivers (e.g. Irish Shelf) is also positively correlated with the NAO and shows persistence beyond the winter season. The regions which include much of the European coast (Armorican Shelf, English Channel, and southern North Sea) show little correlation of runoff with the NAO, perhaps reflecting the larger catchments not having time to respond to the NAO, and their more southerly location. The river correlation patterns were consistent with those of Rödel (2006) and Bouwer *et al.* (2008).

Having shown the correlations of the NAO with the important surface forcings, we now look directly at the relationship between the observed DJF NAO and the shelf response (Figure 8). We find a significant positive correlation between the winter NAO and the winter SST in the most southern and eastern shelf regions (Figure 8a), consistent with previous studies (Dippner, 1997; Hurrell and Van Loon, 1997). In most of these regions, the significance of these correlation (at the 95% confidence level) persists for one month (Figure 8b), and in the English Channel and southern North Sea a second month (February-April (FMA) SST, Figure 8c). The NBT correlations also show significant correlations with the NAO and having memory in some regions. The DJF NAO is generally not significantly correlated with SSS (Sea-Surface Salinity) in most regions, although there is a significant correlation in the Skagerrak/Kattegat, which persists until FMA (Figure 8g-i). There could be a much lower frequency salinity response to the NAO (e.g. Belkin et al., 1998; Mysak et al., 1996), which would not be captured by these correlations. Such a response may provide predictability on longer time scales.

The above results suggest that knowledge of the NAO index could provide some skill for important variables, at modest lead times of 1-2 months, even if the DJF NAO is only determined from observations (at the end of the December-February period). Because the GloSea5 system has skill in predicting the DJF NAO index from the previous November (Scaife et al., 2014), it is possible that the lead time could be increased by using the predicted rather than the observed NAO index. In Figure 9 we examine correlations for the same predicted NWS variables as in Figure 8, but this time using the DJF NAO index predicted from the ensemble mean of the GloSea5 forecast run the preceding November. Unsurprisingly the correlations are generally lower than for the observed NAO index, and many are not statistically significant at the 95 % confidence level. However, the correlations are largely of the same sign and pattern as for the observed NAO index. This suggests that a prototype seasonal forecast based on the GloSea5 NAO may be possible. For SST and NBT, there are regions with exhibit persisting significant correlations (e.g. English Channel) which is promising. The correlation patterns for SSS

are however, quite different to those from the observed NAO. There is a general negative correlation across the shelf that gets stronger with an increasing lag (Figure 9g-i). The persistence in the NAO-SSS correlation reflects the longer term nature of salinity anomalies. The difference in the SSS correlation patterns between the observed and GloSea5 NAO perhaps act as an error estimate to this approach, suggesting caution and further assessment is needed before relying on an empirical seasonal forecast of this form for SSS. Overall, the results with the GloSea5 NAO suggest that real relationships exist between the forecast NAO and the observed NWS fields, and that further improvements in the seasonal NAO forecast would deliver higher levels of forecast skill and/or regional detail.

The correlations between the NAO and the NWS fields describe how strong a linear relationship exists between the two. Where there is significant skill (a significant correlation, Figure 8) this linear relationship can be used to predict the NWS fields from the NAO. A simple approach would be to find the slope and intercept between the observed (NOAA) NAO and the shelf seas variable, and then apply this equation to the GloSea5 forecast NAO. This provides a simple empirical forecast giving information about the future state of the NWS (e.g. greater than average, less than average). Other non-linear relationships (e.g. quadratic etc.) may exist between the NAO and NWS fields that could be used as the basis of an empirical forecast – further analysis (and curve-fitting) would be required for their identification. The reanalysis provides a coherent dataset in order to explore such relationships.

Here we present such an example forecast of the English Channel SST initialised in November, for the following winter. Using the reanalysis, we construct a linear empirical relationship between the observed (NOAA) DJF NAO and the English Channel winter (DJF) SST. The correlation in Figure 8 quantifies the strength of this linear relationship, and its significance. We then apply this equation to the GloSea5 forecast NAO (Figure 10). Due to the persistence of the NAO SST correlation in this region, we are able repeat the process to extend the lead time (February-April forecast initialised in November), beyond which the underlying correlations significance is greater than 0.05. These (normalised) forecasts are illustrated in Figure 10. While our example has focused on a region and variable with a relatively high correlation, the English Channel winter SST may have a direct application. For example, European Sea Bass (*Dicentrarchus labrax*) spawn between the southern North Sea and the Celtic Sea, in February-April within the 9 °C isotherm – this region expands in warmer years (Beraud et al., 2017). Sea bass is a high value fish that is exploited by commercial fisheries (ICES, 2012) and is an important species for recreational anglers. The English Channel SST forecast for February-April (FMA, Figure 10) would be directly applicable to Sea Bass spawning. Furthermore, as the GloSea5 based forecasts also have skill, these forecast may be made in November (when the GloSea5 DJF NAO forecasts are made), and so provide February-April forecasts as early as November, such results could be used to inform precautionary management when needed. The equations for these prototype forecasts are given in Figure 10.

Developing a range of empirical forecasts is a possible way of producing NWS seasonal-forecasts, especially if based on predictable climatic indices such as the NAO. Investigating the relationships between the climatic drivers and the shelf response within NWS climate control simulation (i.e. using a multi-century global climate model run with fixed climate forcings, to drive a multi-century NWS simulation (Tinker et al., in preparation)) may allow much subtler relationships to be established than is possible with the relatively short modern observed period. However, due to empirical forecast's reliance of past observations, there can be limits to their use in the future, where conditions are outside the present day range.

### 3.4. Question 4: What are the prospects for improving NWS seasonal forecasts?

With the maturity of seasonal forecasting systems, and shelf seas dynamic downscaling systems, it will not be long before a seasonal forecast system for the NWS, based on dynamical downscaling is technically possible. The underlying skill of the global seasonal forecasting system would be the basis to any such NWS downscaling, and its skill, resolution etc. will be the leading order limit to the subsequent skill of the NWS seasonal forecast. However, the skill from the global seasonal forecasting system must be able to propagate from its ocean and atmosphere and manifest itself on the NWS for a NWS

forecast to have any skill. We can start to explore whether this is the case with the CMEMS reanalysis. If there is only a weak relationship between the boundary (lateral and surface) and the interior of the shelf, and internally generated chaotic variability dominates, any year-to-year variability modelled by GloSea5 will not manifest in the NWS. The NWS is considered to be quasi-isolated from the North Atlantic (Wakelin et al., 2009), however, it is a broad continental shelf sea and so interaction with the atmosphere is important. We now investigate the relationships between the state of the NWS in the CMEMS reanalysis and the boundary conditions that forced it, to see how much of the NWS variability is driven by the large-scale drivers, and how much is internally generated. We focus on the surface boundary conditions, as the advective lag between the lateral boundaries and the shelf make them less relevant for interannual variability and seasonal forecasting.

We consider the inter-annual correlations between the CO5 NWS reanalysis surface and open ocean boundary forcings and the NWS temperature and salinity in the regions defined in Figure 1 (see Figure 11). Our interpretation of the correlations must be informed by physical insight, recognising that correlation does not imply causation. We examine processes influencing temperature and salinity separately, since temperature is likely to be strongly influenced by surface heat exchange, whereas advective processes may play a stronger role for salinity due to a lack of direct feedback between surface salinity and the atmosphere.

NWS surface temperatures are significantly correlated with temperatures at the open ocean boundaries of most of the domain (the northern boundary west of 10° W being the exception (not shown); example of other ocean boundary sections shown in Figure 11a, d). Water advected from the lateral boundaries transports heat and salt onto the shelf, which influences these correlations. Heat is exchanged with the atmosphere faster than freshwater, so the memory of temperature of the lateral boundaries is overwhelmed by the surface heat exchange, while salinity memory persists longer. Therefore, the shelf SST correlations with the oceanic temperature boundary conditions is likely to be due to common surface forcings acting on the open ocean and shelf seas temperatures. Shelf SST is strongly correlated with surface air temperature as expected with the use of CORE bulk formulae. This is very homogenous across the shelf, so that the annual mean surface air temperature over the central North Sea is significantly correlated with the SST in all shelf regions (Figure 11g). This is also true of the humidity (Q2, Figure 11h) and the downward component of the thermal radiation (STRD, Figure 11i). The incoming solar radiation (SSRD) has smaller spatial scales, consistent with the synoptic spatial scale of the atmosphere (Figure 11j-l): in the annual mean, there is a positive correlation between incoming solar radiation in an example region (Northern North Sea) and the SST in the same region (Figure 11l), however the correlations reduce with distance from the region. There is also a strong seasonal cycle in the incoming solar radiation correlations (Figure 11k, l) with strong positive correlation when comparing summer incoming solar radiation and SST (clear sunny summer days imply strong solar heating) and (insignificant) negative correlation in the winter (clear sunny winter days are associated with strong long-wave night time cooling). The use of the CORE bulk formulae forcings preclude the investigation of the net radiative fluxes, as the outgoing radiation is calculated by the ocean model rather than being prescribed. Overall, variability in SST appears to be linked to large-scale drivers (which may be predictable), with some contribution from less-predictable synoptic scale variability at the regional scale.

The salinity on the NWS is primarily a balance between salty water entering the shelf from the North Atlantic, and its modification due to water exchanges with the atmosphere (e.g. precipitation) and dilution from rivers (and exchange with the Baltic). This leads to some intuitive relationships: the saltier the Atlantic is, the saltier the NWS; the greater the river flow and rainfall into the NWS, the fresher the NWS is. These can be considered the direct mechanisms that controls salinity. An important secondary mechanism is the rate at which the NWS water is exchanged with the Atlantic.

The salinity in the regions of the CO5 NWS reanalysis domain off the shelf is strongly correlated with the salinity of the boundaries. For example the salinity in the north-western oceanic part of the domain is strongly correlated with the salinity boundary forcings along the western edge of the domain (e.g. Western Boundary - Surface_Central is correlated with SSS in Figure 11b) and this penetrates onto the shelf in the Shetland shelf region. The salinity of the large south-western oceanic

region is positively (though not significantly) correlated with the surface salinity of the southern boundary (Southern Boundary - Surface_East), but there is a suggestion that the influence of boundary forcing is advected onto the shelf through the Armorican Shelf, Celtic Sea, English Channel and into the southern and central North Sea due to significant correlations (Figure 11e). Note exemplar ocean boundary conditions in Figure 11a-b,d-e are not necessarily the strongest correlations between the shelf and the ocean boundary conditions, but have been chosen to provide a pair of regions to highlight the west/south temperature/salinity differences in correlations. The lack of significance in the correlations between the SW oceanic region and the southern boundary salinity may reflect this large region blurring out a smaller correlated area (i.e. the tightly defined European slope current flowing northwards from the boundary towards the NWS).

There can be considerable advective lags between the ocean/lateral ocean boundaries, and the NWS. Analysis of Holt et al. (2012) suggests modelled conservative tracers in the open ocean took ~5 years to propagate on the NWS (when the shelf concentration reach ~80% of the ocean – their Figure 8). High-frequency salinity anomalies at the lateral boundaries will not be correlated with NWS SSS annual means (i.e. Figure 11), but lower-frequency anomalies may be. As well as the instantaneous correlations between the shelf and lateral boundary annual mean salinities, we have also investigated the maximum lagged correlations (not shown). We found that the lags tend to corroborate Holt et al. (2012), with largely significant correlations between most shelf regions and the north and western boundary with lags of ~5-6 years, and significant correlations between the southern regions of the shelf and the southern boundary with 0 year lag (Figure 11c). This reflects the different lags for the different advective pathways. These lag correlations of 5-year between the lateral boundary conditions and the NWS SSS may suggest predictability at decadal time scales, but were not further considered here (due to study scope, and the reanalysis length). Within a seasonal prediction system, they would be included via the initial conditions.

Runoff from large rivers tends to be (negatively) correlated with salinity across the shelf (e.g. the European rivers that flow into the southern North Seas region; Figure 11f), while runoff into regions that only include smaller (e.g. UK and Irish) rivers tends to have only local effects (such as the rivers flowing in to the Irish Sea; Figure 11c).

The salinity across the shelf is significantly negatively correlated with total precipitation in the south of the domain (Figure 11p), but not with precipitation in the north of the domain (Figure 11m), which is insignificantly positively correlated. Conversely, strong wind magnitudes in the north of the domain are correlated with salinity across the domain (Figure 11o), but wind magnitude in the south of the domain is not (Figure 11r). Mean Sea Level Pressure (MSLP) in the south of the domain is positively correlated with salinity across the shelf (Figure 11q), but northern mean sea-level pressure is not (Figure 11n). These patterns suggest that the NWS salinity is responding to a large-scale driver (rather than simply local rainfall/river flow) such as varying exchange with the Atlantic. The meteorological relationship between these surface forcings (low mean sea-level pressure, high wind and rain) suggests stormy conditions, and the North/South correlation patterns of mean sea-level pressure, wind magnitude and total precipitation suggest storm track latitude may be important.

When we compare these variables against the winter storm track latitude (Figure 12), we find that a more northern storm track location does correlate negatively (positively) with mean sea-level pressure (wind magnitude and total precipitation) in the north of the domain, and the opposite in the south. We find that these correlations tend to reduce strength through the seasons being highest in winter (DJF) and reducing through to the spring (MAM). This suggests that the index of storm track latitude is important for these variables.

We find that shelf salinity is (insignificantly) positively correlated with a more northerly storm track, particularly in the Irish and Shetland Shelf, and the northern and central North Sea, and there is a suggestion of a lag in the correlation as you move towards the North Sea. One possible mechanism that could explain these correlations is the rate of exchange of water with the Atlantic.

Exchange of water between the NWS and the Atlantic tends to dominate the mean salinity of the North Sea (Sündermann and Pohlmann, 2011). One of the main pathways of water into the North Sea is via the Fair Isle Current (Sheehan et al.,

2017), fed by the European Slope Current. Both the Fair Isle Current and the European Slope Current have been shown to correlate with the NAO (Marsh et al., 2017; Sheehan et al., 2017; Winther and Johannessen, 2006). This positive correlation means more (relatively) high salinity is advected onto the NWS under NAO positive condition, which would tend to increase the NWS salinity. We do not explicitly look at the European Slope Current in this study, but understanding its predictability could provide an important mechanism for NWS salinity predictability.

Because the strength of the inflow of North Atlantic water onto the shelf (the indirect mechanism affecting the year-to-year shelf salinity) and the direct mechanism (dilution from rivers and precipitation) are both correlated to NAO, but in an opposing manner, it is possible more information (than is contained within the simple NAO index) is needed to predict NWS salinity variability. Such a balance of opposing mechanisms may explain the relatively low NAO SSS correlations (Figure 8 g-i) despite the main drivers of salinity being correlated to NAO. In principle, dynamic downscaling may provide additional skill by modelling this mechanism directly, although empirical forecasts can be designed to capture the opposing mechanisms.

The spatial patterns in the relationships in Figure 11 show that (some of) the variability on the NWS is strongly coupled to the large-scale boundary conditions – this is consistent with the established view that the NWS is a boundary driven system (with the caveat that we are using a non-eddy permitting model for the NWS). Hence the overall concept of (empirical or dynamic) down-scaling based on large-scale boundary drivers that may be predictable by a global seasonal forecast system is plausible. This suggests that a key area for scientific effort is to evaluate and improve the predictability of the NWS boundary drivers as produced by GloSea5.

Overall, we conclude that much of the year-to-year NWS variability is relatively tightly linked to the variability in the boundary conditions, which is a prerequisite for dynamic downscaling. We note that in some cases, there may be a balance of opposing mechanisms, or response to a sequence of NAO events, that may require more information that is encapsulated in the simple NAO Index - this may provide a pathway for additional predictability from dynamic downscaling when compared to empirical downscaling. Furthermore, much of the temperature and salinity variability on the NWS is linked to large-scale climate variations (including river outflow which integrates rainfall over a large area) rather than to more local effects (such as the direct effect of rainfall on the synoptic scale). This increases the prospect of useful seasonal predictions since global seasonal forecast systems are beginning to show significant skill in predicting large-scale climate indices.

## 4. Conclusions and Prospects

Our preliminary investigation shows that despite the useful skill that GloSea5 has in predicting certain large-scale climate indices, its output may limited direct application for shelf seas seasonal forecasts because of limited resolution, missing shelf sea processes and simplified treatment of river runoff. However, we have shown evidence that many aspects of inter-annual variability on the NWS are driven by large-scale variations in elements of the atmospheric, oceanic and riverine forcings that are closely linked to the winter NAO index, for which there is considerable predictive skill at a lead time of several months. Indeed Figure 10 shows that a simple empirical downscaling approach driven by the forecast NAO index can provide significant skill in some variables/regions at a lead time of several months. Further improvements to the reliability of the GloSea5 NAO forecast may enhance the forecasts for the NWS region. While being skilful, GloSea5 has been shown to be under-confident (meaning there is too high a proportion of unpredictable noise in the forecast) for the NAO and the wider Atlantic region (Eade et al., 2014), and for the inter-annual predictions (Dunstone et al., 2016). This problem is common to most skilful seasonal forecasting systems (Baker et al., 2018), but as there are exceptions, it is not inherent to such systems. Solving the under-confidence issue is an active research area.

Based on our results we can make an assessment of the three possible forecasting methods identified in the introduction:

A. Read off the NWS marine variables directly from the underlying climate model. We have shown that this is not feasible under some (e.g. stratified) conditions with current generation seasonal forecast systems. However, more

research is required to identify if and when this approach may be appropriate. Over the coming 5-10 years it is expected that such global seasonal forecasting systems will move to higher ocean resolution, and may incorporate tidal processes and improved coupling with river hydrology (e.g. Holt et al., 2017). Hence this approach may become feasible in time, although the resolution of global climate models will remain coarse compared with what is achievable through regional models.

B.  Empirical downscaling. Our results show that a significant level of skill can be achieved for a limited set of variables at a few months' lead time. The very limited availability of long observed time-series on the NWS means that the empirical climate response functions will often need to be developed using reanalyses. Our evaluation of the CO5 NWS reanalysis has shown encouraging evidence of its ability to capture inter-annual variability.

C.  Dynamic downscaling. For the NWS variables that cannot be skilfully forecast directly from the NAO, additional predictability might be possible by dynamic downscaling. With the dynamic downscaling approach, much more information from the global seasonal forecasting system (including both predictable and unpredictable components) is used. This may allow important subtleties that are not captured in a simple NAO index, to be resolved. Additionally, persistence (encapsulated in the initial conditions of the regional model) can provide additional skill. Our analysis of the relationship between the NWS variability with the boundary drivers corroborates the boundary constrained nature of the NWS (e.g. Holt et al., 2016), and so supports the possibility of additional forecast skill from dynamical downscaling, provided the driving global system can forecast the driving boundary conditions reliably. However, this approach requires significant additional research.

Even the limited level of predictive skill we have shown here for some regions of the NWS may be useful for certain applications, e.g. SST forecasts for February-April may give early indications of increased risk of harmful algal blooms, and predictions of near bottom temperature and its impacts on the gas supply network may inform more resilient energy planning. Further developments will be needed to deliver a seasonal prediction system for the NWS with sufficient skill and reliability to inform user planning decisions over a wide range of applications. Specific research priorities are:

- Assess user value for cases where we already have demonstrated some skill, and establish what skill/reliability would be needed to provide actionable forecast information in various sectors
- In-depth assessment of regional reanalyses as a tool to develop empirical downscaling relationships
- Assess if and when/where the GloSea5 seasonal forecast system can be used directly for NWS forecasts.
- Assess potential added value of dynamical downscaling, initially through case studies
- Identify the largest sources of uncertainty in downscaled predictions (e.g. seasonal forecast fields for specific drivers, downscaling model), to inform where to focus development effort
- Assess predictability in seasons other than winter. To date seasonal forecasting systems have shown less skill in the summer, but this is an active research area (e.g. Hall et al., 2017). It may also be possible to demonstrate some degree of memory in the shelf seas themselves, which would add to forecast lead times.

Many challenges remain before we can derive seasonal forecasts for the NWS that are accurate and reliable for a wide range of regions and variables. It is likely that some variables will prove to be inherently unpredictable to any useful degree. But the early results presented here show that current seasonal forecast systems can already provide meaningful information with the potential for applications in marine operations and planning. As our understanding and capability develops, a close interaction between climate scientists, marine scientists and end users will be needed to bring the added value of seasonal forecast information into decision making in marine policy, planning and operations.

## 5. Data availability

The data used in this study is available from the Copernicus Marine Environment Monitoring Service (CMEMS) Northwest European Shelf reanalysis (NORTHWESTSHELF_REANALYSIS_PHYS_004_009) available from their online catalogue.

## 6. Author contribution

JT designed the analysis, which was undertaken by JK and JT. JT and RW prepared the manuscript with contributions from all co-authors. RB, SD provided advice on observations datasets, policy and user relevance.

## 7. Competing Interests

The authors declare they have no conflict of interest.

## 8. Acknowledgements

The authors thank Leon Hermanson, Craig MacLachlan and Drew Peterson (MOHC) for advice and support with working with GloSea5, and John Pinnegar (CEFAS) for his comments and proofreading, the reviewers for questions and comments that strengthened the manuscript (particularly relating to the reanalysis evaluation), and Enda O'Dea, Dave Storkey and Robert King (MOHC) for Nemo advice. This work was supported by the Joint BEIS/Defra Met Office Hadley Centre Climate Programme (GA01101) and by Defra ME5317 ForeDec – Developing the capability to use marine forecasts over seasons to decades. The Southern North Sea Ferry time-series work has been supported by Defra ME Cefas- SLA15/25 and is a contribution to the ICES Report on Ocean Climate (https://ocean.ices.dk/iroc/). The Western Channel Observatory E1 time-series is maintained by the Plymouth Marine Laboratory and the Marine Biological Association, and is supported by UK's Natural Environmental Research Council's National Capability programme. The SST time-series from the Malin Head meteorological station was supplied by the Marine Institute and Met Éireann in Ireland. The temperature and salinity time-series from the Fair Isle was supplied by the Marine Scotland Science. The temperature and salinity time-series from the Helgoland Roads was supplied by the Alfred Weneger Institut/Helmholtz-Zentrum für Polar und Meeresforschung. This publication has been developed in cooperation with the European Union's Horizon 2020 research and innovation project AtlantOS (633211).

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

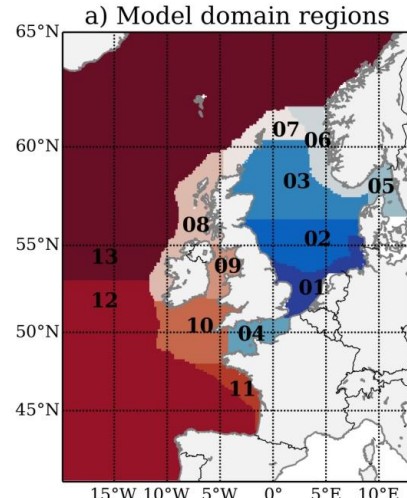

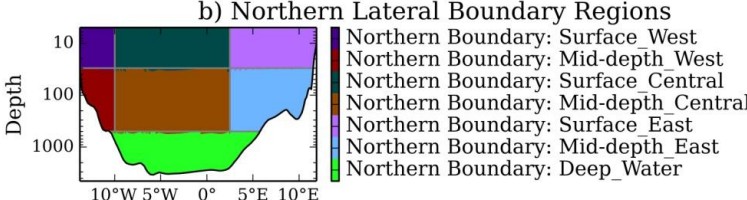

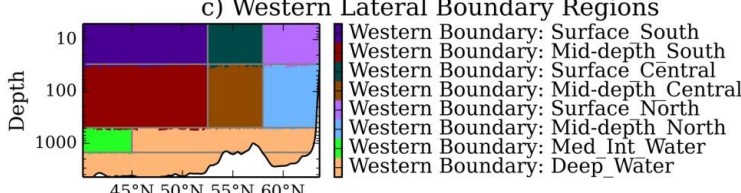

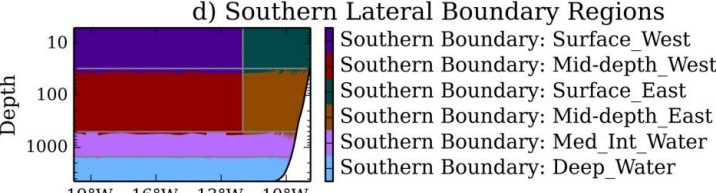

Figure 1 a) CO5 NWS reanalysis domain with coloured analysis regions , numbered as: 01 Southern North Sea; 02 Central North Sea; 03 Northern North Sea; 04 English Channel; 05 Skagerrak/Kattegat; 06 Norwegian Trench; 07 Shetland Shelf; 08 Irish Shelf; 09 Irish Sea; 10 Celtic Sea; 11 Armorican Shelf; 12 NE Atlantic (S); 13 NE Atlantic (N). b-d) Region mask used for the oceanic lateral boundary condition: b) The Northern Boundary is divided at 10° W and 2.5° E, and at 30 m and 500 m (below 500 m the boundary is not separated by longitude); c) The Western Boundary is separated at 30 m and 500 m, and at 52.5° N and 58° N. Mediterranean Intermediate Water is identified between 500 m and 1500 m south of 45° N, and this is separated. d) The Southern Boundary is divided at 30 m, 500 m and 1500 m. Above 500 m the boundary is separated at 12° W.

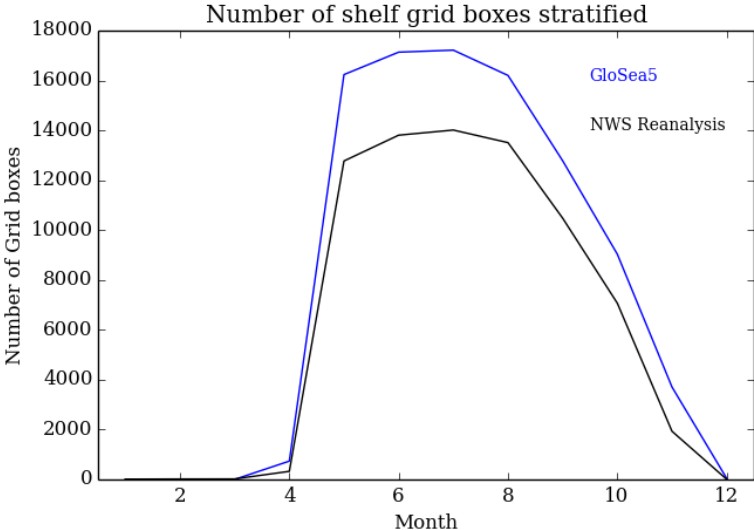

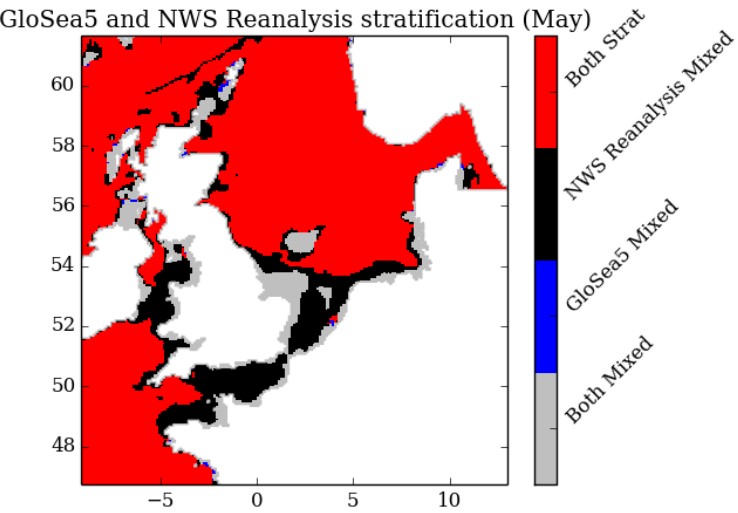

**Figure 2 a) 20-year mean seasonal cycle of stratified extent for CO5 NWS reanalysis (black) and GloSea5 (blue). b) 20-year mean stratification map (for May, where (SST – NBT) = 0.5 °C). Grey and red denote regions where both models agree that there is a mixed and stratified water column respectively. Black and Blue denote regions where the CO5 NWS reanalysis and GloSea5 disagree – black being where NWS considers the water column to be mixed, while GloSea5 considers it to be mixed, and blue being the opposite.**

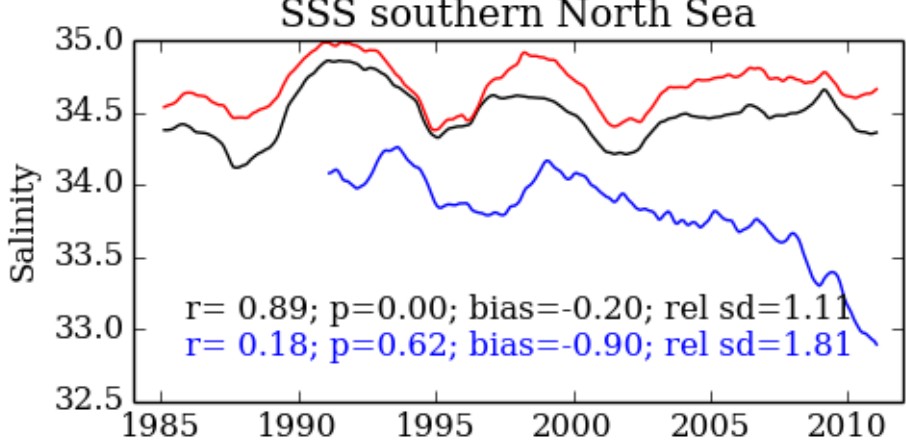

**Figure 3 Time-series of observed (ferry samples) southern North Sea salinity (red) compared to CO5 NWS reanalysis (black) and GloSea5 (blue) data, as 2-year running mean. Both modelled time-series are from the nearest model grid-box to the observations.**

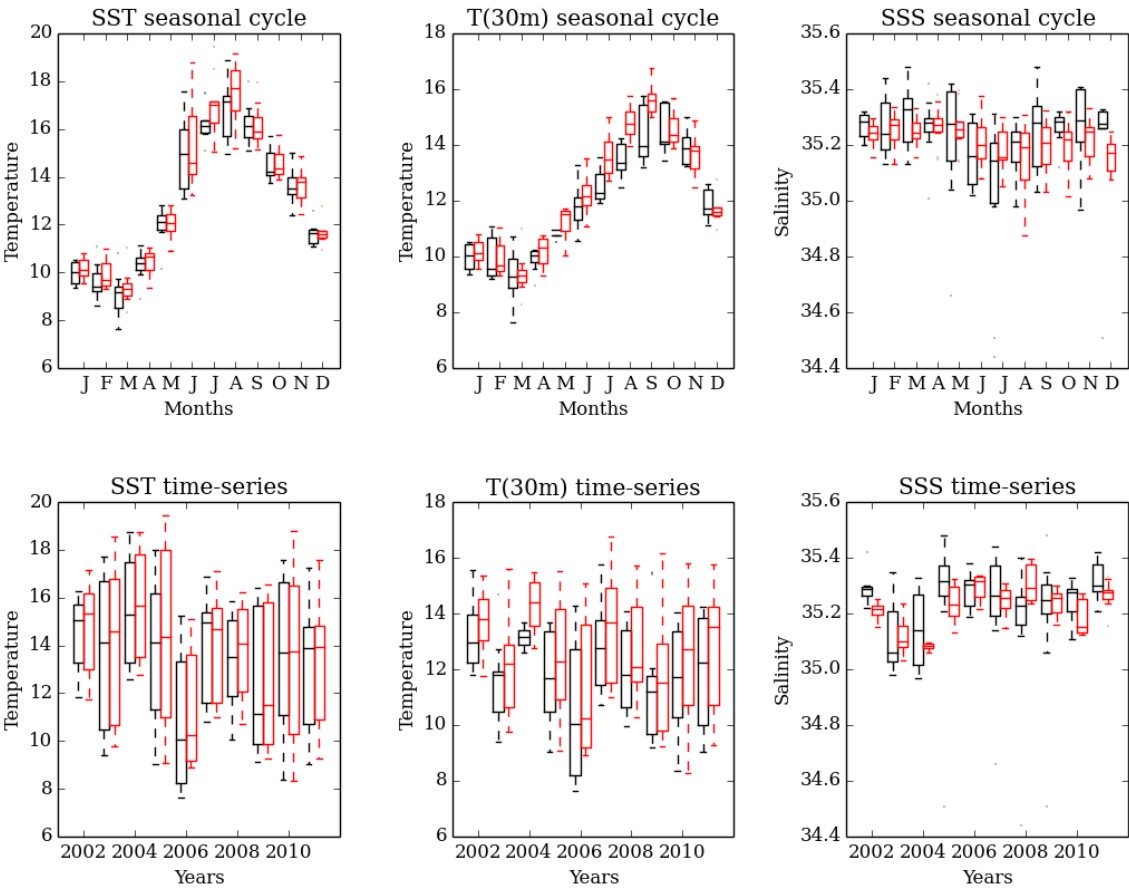

**Figure 4 WCO E1 observed (black) temperature (surface: SST; and 30 m: T(30m)) and salinity (surface: SSS) compared to CO5 NWS reanalysis (red), showing the mean seasonal cycle and inter-annual time-series (including both winter and summer observations). The nearest CO5 NWS reanalysis gridbox to the WCO site is used, without horizontal or vertical interpolation.**

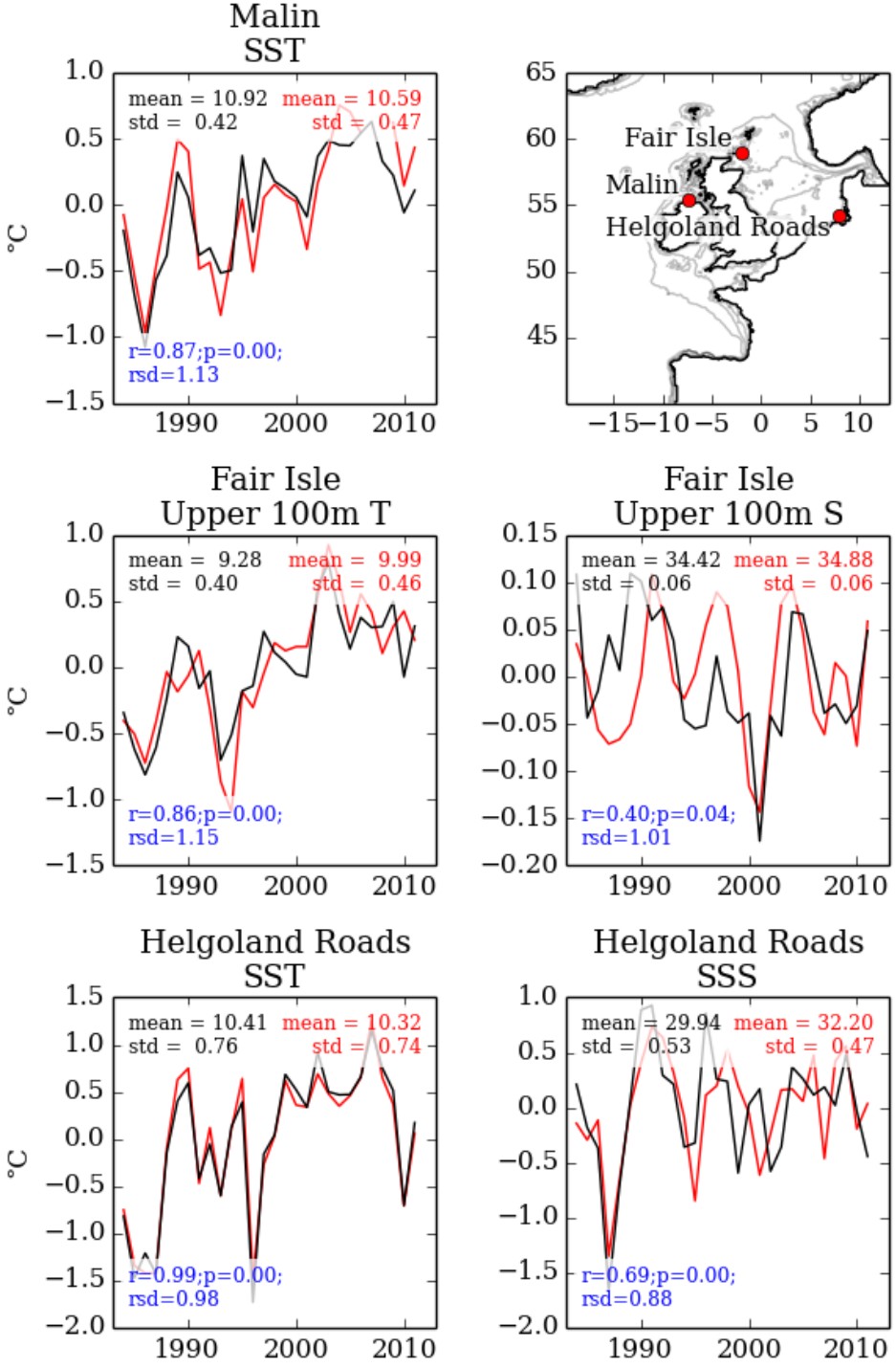

**Figure 5 Reanalysis evaluation with ICES temperature and salinity data. Data is presented as anomalies (the time-series mean is removed) to highlight the variability (reanalysis red; observations black), however the mean and standard deviation (and Pearson's correlation coefficient (r, with its significance (p)), relative standard deviation (rsd), and root mean square) are given. Temperature (left panels) and salinity (right panels) from three ICES time series: Malin Head (upper left); Fair Isle (middle row) and Helgoland Roads (lower row). Upper right panel shows the location of these time-series (the Malin Head dataset does not include salinity).**

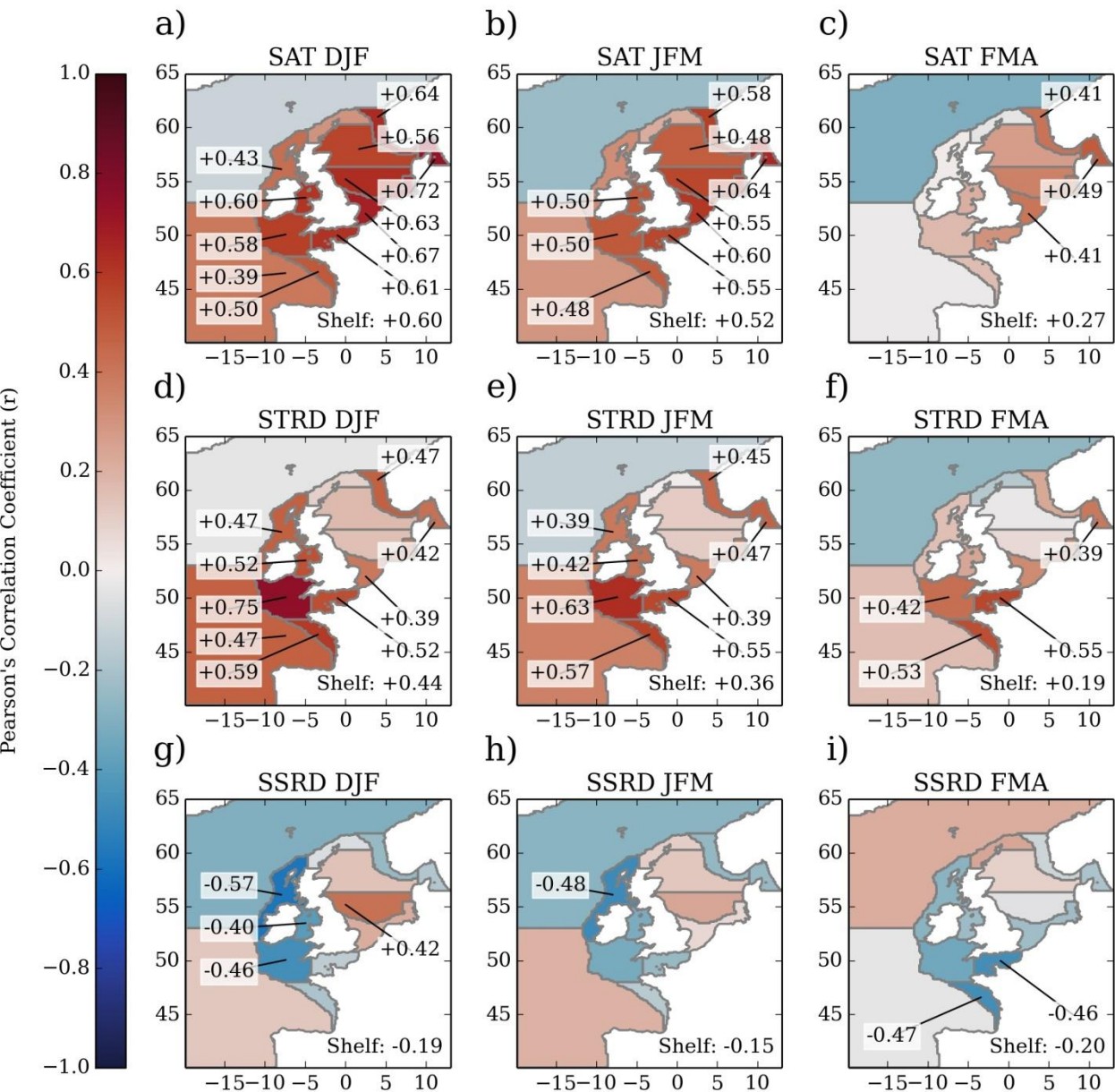

**Figure 6 Correlation maps between winter (DJF) NAO and atmospheric surface fields associated with shelf temperature (Surface Air Temperature (SAT; a-c), Downwards component of the Thermal Radiation at the surface (STRD; d-f) and Downwards component of the Short-wave (Solar) Radiation (SSRD; g-i), upper to lower rows respectively). The left hand panels denote zero-lag (a, d, g), correlating the winter (DJF) NAO with the winter (DJF) surface forcing. The following columns denote 1 and 2 month lags, correlating the winter (DJF) NAO against the JFM (Jan-Mar; b, e, h), FMA (Feb-Apr; c, f, i) surface forcing respectively. The correlation values are given for the shelf, and for regions where the correlation is significant at the 95 % confidence level.**

# Surface Forcing lag corr with NOAA DJF NAO (1985 - 2011)

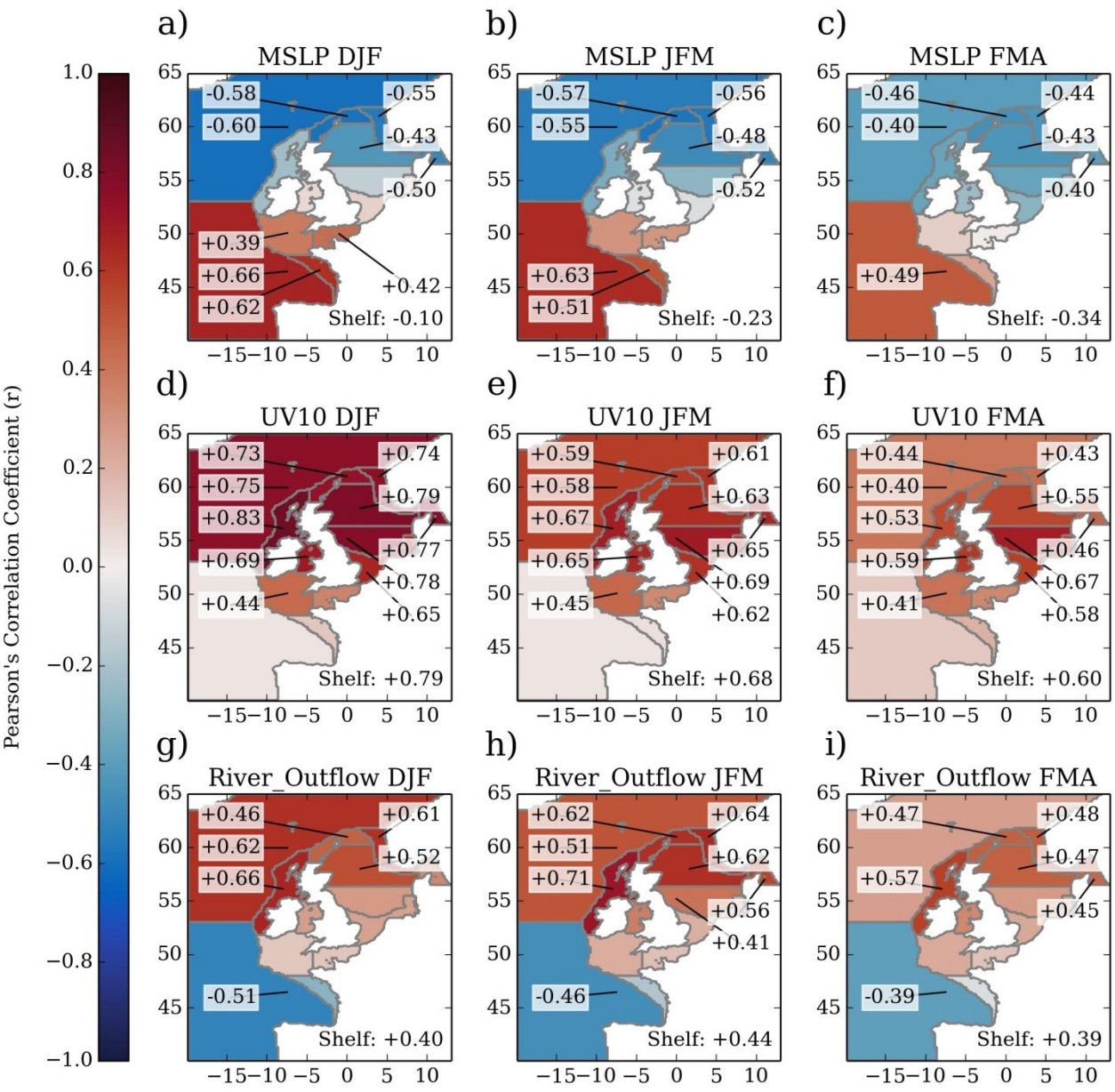

**Figure 7 As Figure 6, but for atmospheric surface fields associated with SSS variability: Mean Sea Level Pressure (MSLP; a-c), wind magnitude (UV10; d-f) and river outflow (g-i).**

# SST, NBT, SSS and SAT lag corr with NOAA DJF NAO (1993 - 2011)

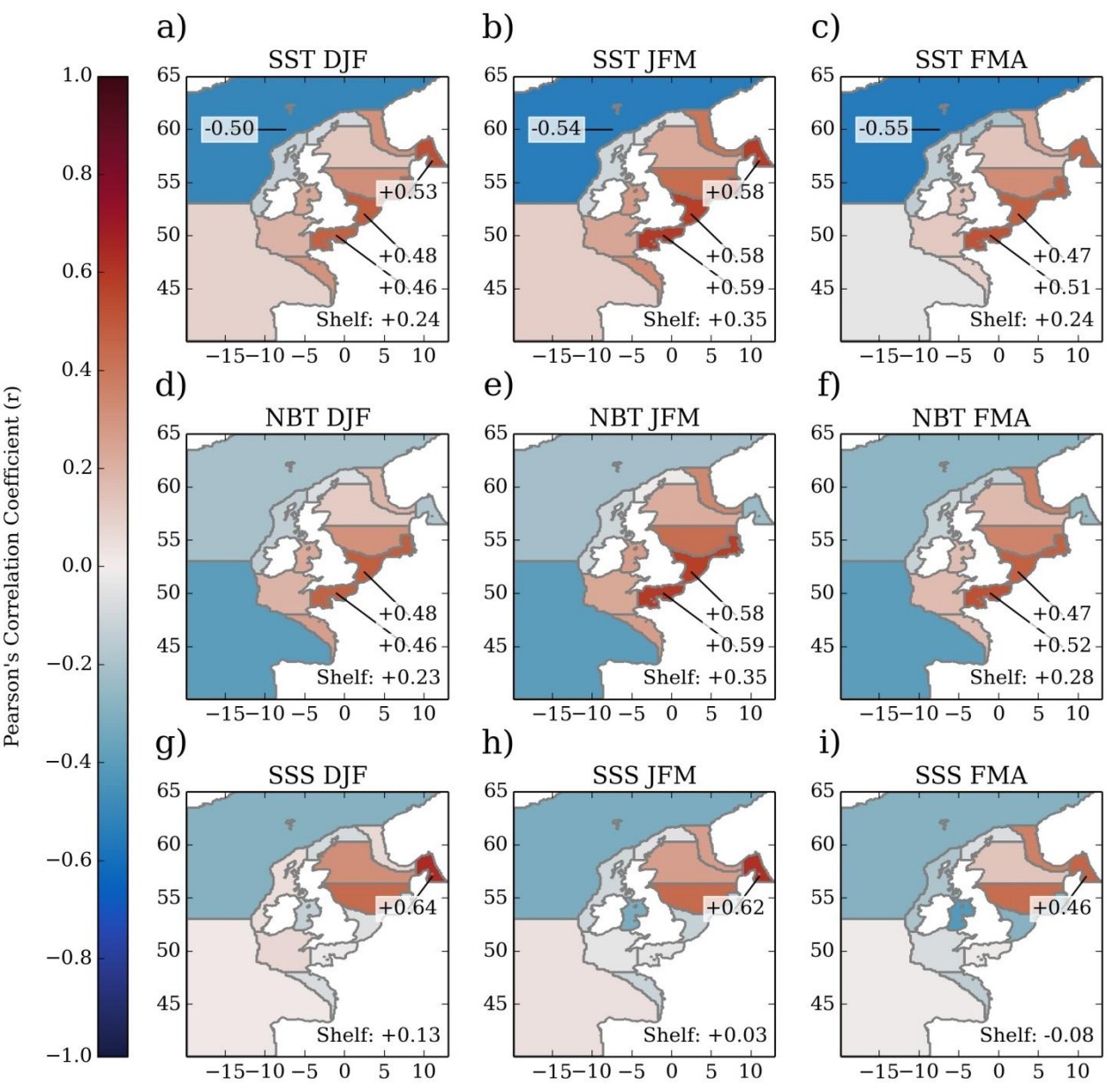

**Figure 8 As Figure 6, but for the shelf response itself: SST (a-c), NBT (d-f) and SSS (g-i).**

# SST, NBT, SSS and SAT lag corr with GloSea5 DJF NAO (1993 - 2011)

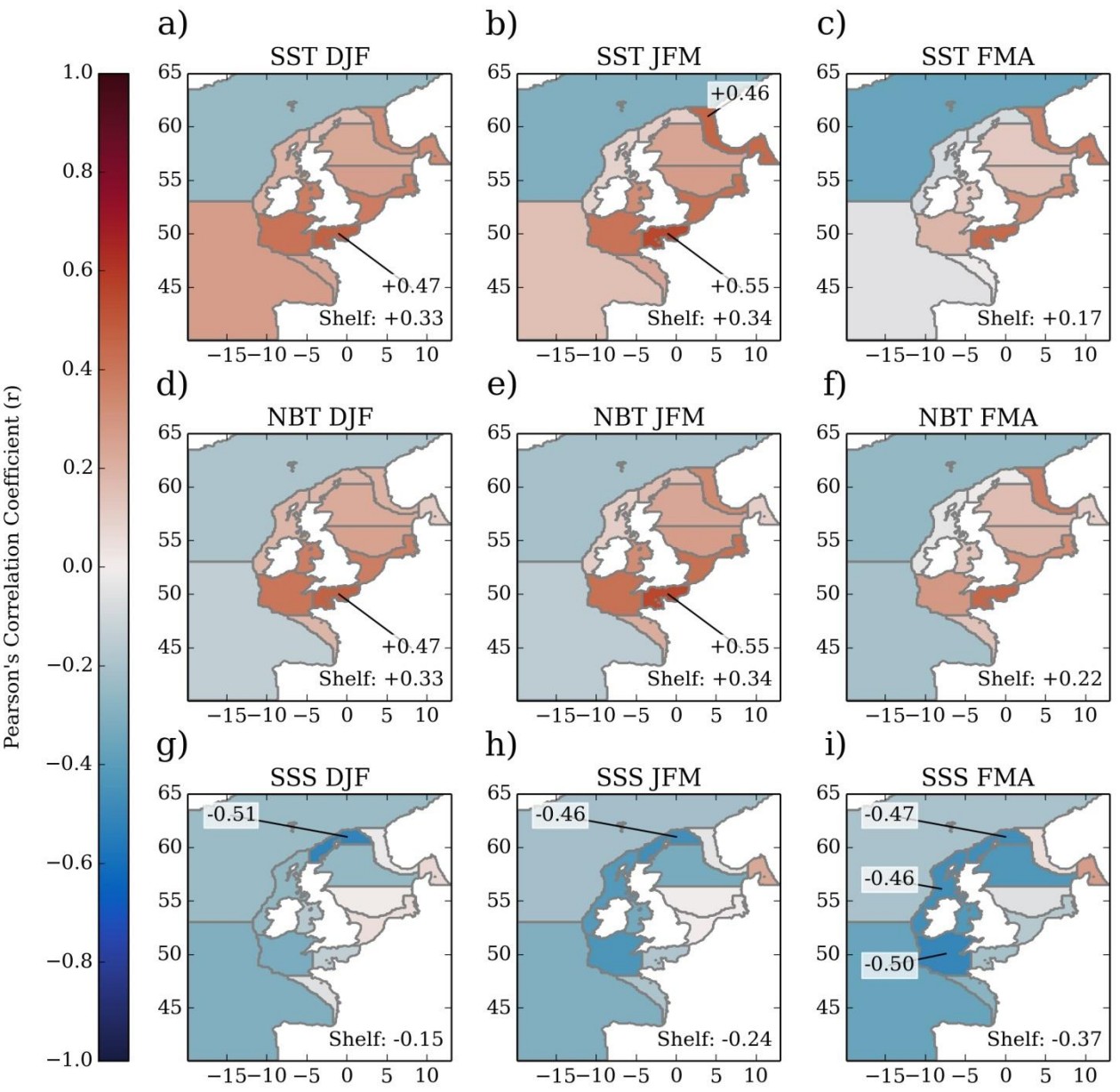

**Figure 9 As with Figure 7, but using the GloSea5 forecast DJF NAO: SST (a-c), NBT (d-f) and SSS (g-i).**

## Prototype seasonal forecast
## English Channel SST
## Reanalysis vs NAO based forecast

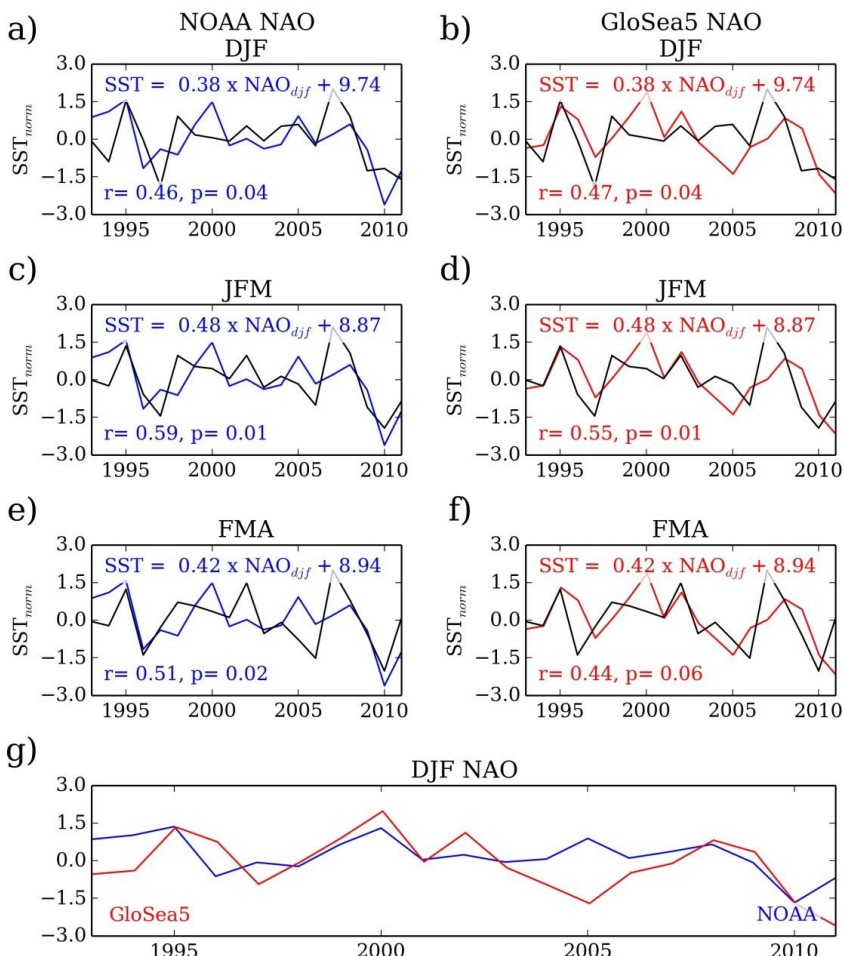

**Figure 10 Exemplar regional seasonal forecast of English Channel SST (selected for the strength of the correlations): The seasonal forecasts is based on the linear relationship between the reanalysis SST, and the observed NAO (blue, left column: a, c, e) which is then applied to the GloSea5 Forecast NAO (red, second column, b, d, f). Each forecast (coloured line) is based on the DJF NAO, with each row showing the consecutive month (first column: DJF forecast; a, b; second row: JFM forecast; c, d; third row: FMA forecast; e, f). For each panel, the black line represents the NWS reanalysis SST, which is the same for both the obsevered and GloSea5 NAO plots for that row. The coloured line represents the DJF NAO based forecast – while the amplitude of this varies with lag (due to the different equation), the pattern is based on the same DJF NAO, and so the pattern remains the same for each column of a given row. Both modelled and forecast line have been normalised (removing the mean and dividing by the standard deviation) – this does not affect the strength or significance of the correlation. The equation for the forecast is given (before normalisation), with the strength of the correlations in each panel. The lowest panel (g) shows the time-series of the observed (blue) and GloSea5 forecast DJF NAO (red).**

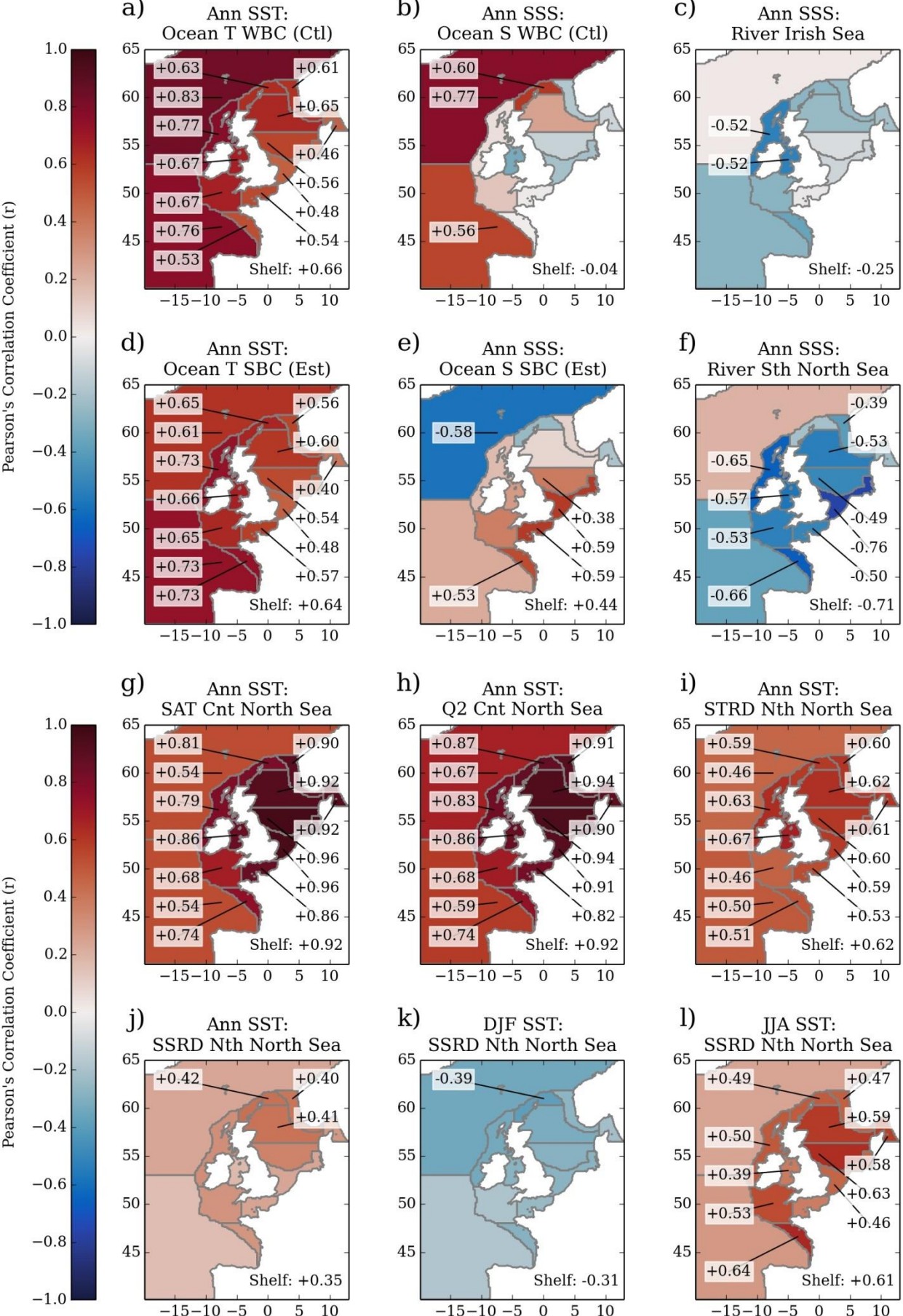

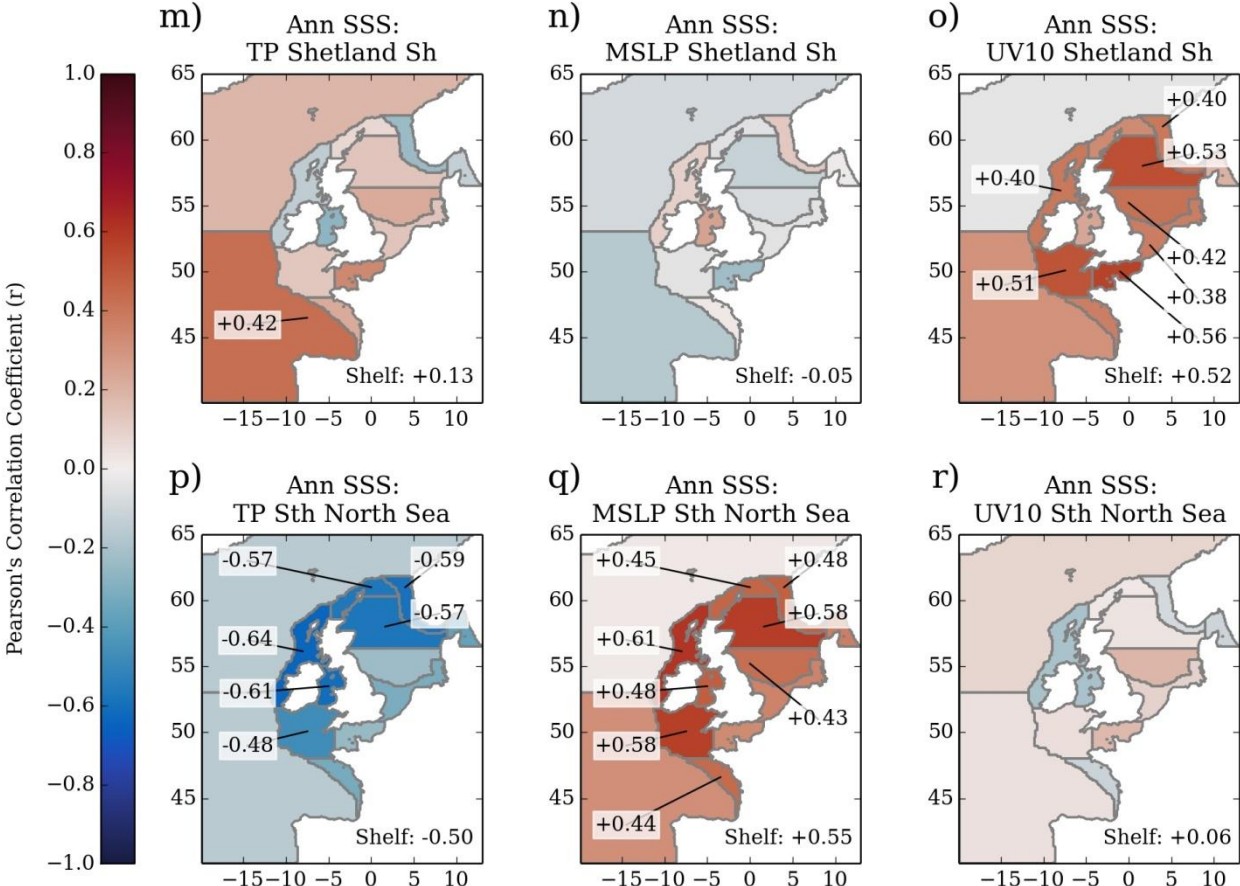

**Figure 11 Correlation maps between time-series (annual mean unless stated otherwise) of model forcings and shelf response (split in 3 sub-panels). Insignificant correlations (at the 95 % confidence level) are not given. The individual panels relate the shelf response to the forcing: a) SST and oceanic temperature at the north of the domain (Western Boundary - Surface_Central); b) SSS and oceanic salinity at the north of the domain (Western Boundary - Surface_Central); c) SSS and river outflow into the Irish Sea; d) SST and oceanic temperature at the south of the domain (Southern Boundary - Surface_East); e) SSS and oceanic salinity at the south of the domain (Southern Boundary - Surface_East); f) SSS and river outflow into the southern North Sea; g) SST and Surface Air Temperature in the central North Sea; h) SST and humidity (Q2) in the central North Sea; i) SST and Surface Thermal Radiation (Downwards; STRD) in the central North Sea; j) SST and Surface Solar Radiation (Downwards; SSRD) in the northern North Sea; k) winter (DJF) SST and winter (DJF) Surface Solar Radiation (Downwards) in the northern North Sea; l) summer (JJA) SST and summer (JJA) Surface Solar Radiation (Downwards) in the northern North Sea; m) SSS and Total Precipitation in the Shetland Shelf region; n) SSS and Mean Sea Level Pressure in the Shetland Shelf region; o) SSS and wind magnitude (UV10) in the Shetland Shelf region; p) SSS and Total Precipitation in the southern North Sea; q) SSS and Mean Sea Level Pressure in the southern North Sea; r) SSS and wind magnitude (UV10) in the southern North Sea. The correlation maps given are not necessarily the strongest correlations, but have been selected to illustrate the observed patterns consistently.**

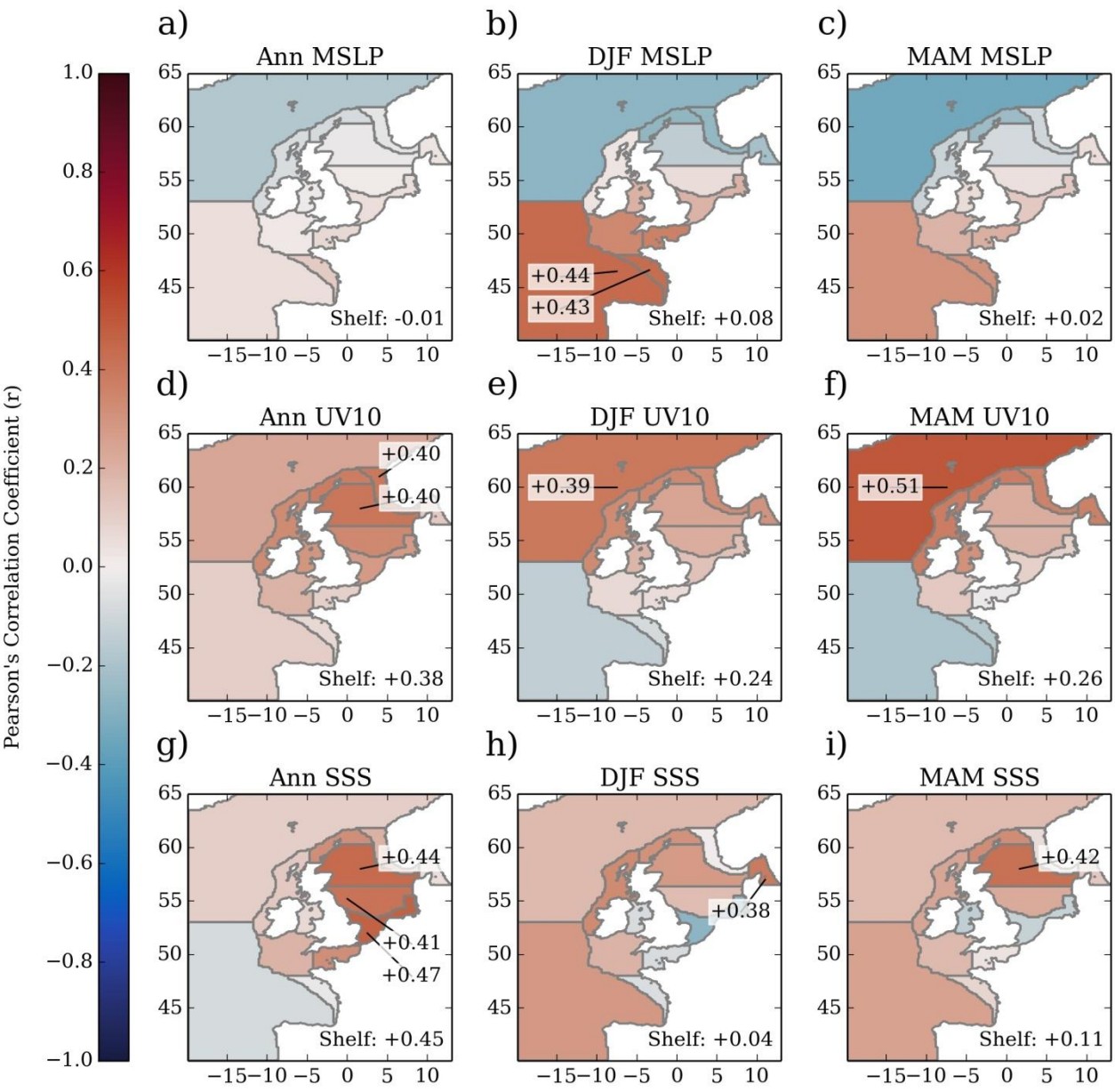

**Figure 12 Correlations between winter (DJF) storm track latitude, and Annual mean, winter and spring (left to right) mean sea level pressure (MSLP; a-c), wind magnitude (UV10; d-f), and surface salinity (SSS; g-i).**