# Peer review of "What are the prospects for seasonal prediction of the marine environment of the Northwest European shelf?"

_Ocean Science, 2018_

## Referee Comment (RC1) · Anonymous Referee #1 · 31 Jan 2018

The presented study by Tinker et al. intends to reflect on three approaches to forecasting physical conditions of the Northwest European shelf (NWS) on monthly to seasonal time scales.

The first approach, that is using a global circulation model (GCM), is rejected by the authors by referring to model limitations of GCMs. However, this rejection is trivial. Since the 1980s the development of regional circulation models (RCM) is motivated by limited representations of shelf sea dynamics in GCMs. Moreover the authors only highlight the deficiency of GCMs in simulating tidal mixing due to the neglect of tidal waves, which is also trivial and known.

Their account on the second approach, that is developing empirical relationships between predictable large-scale driving mechanisms and NWS conditions, is a mere correlation analysis of the NAO with reanalysis atmospheric forcing fields and T and S for various sectors of the NWS and eastern North Atlantic. Moreover, the found correlations are all well known but in the manuscript often not even cited. It is mentioned that correlations do not guarantee causal relations. However, underlying physical mechanisms linking the presented variables are poorly discussed.

Their exemplified "forecast" of English Channel SST based on the NAO index is only a comparison of SST and NAO index time series, quantified by the correlation coefficient. A suggestion how to construct SST from a predicted NAO index, though, is not presented.

The third approach, that is a dynamical downscaling of a GCM forecast by a RCM, is not yet presented but theoretically discussed by correlation analysis of annual mean T and S on the shelf with various boundary forcings for a regional reanalysis product. The idea behind this is that if the boundary forcings are significantly correlated with the NWS sectors and there is forecast skill by the GCM for large-scale driving mechanisms then a dynamical downscaling should be able to improve the forecast for the NWS. However, just as for the second approach, the found correlations are not new and sometimes the correlated variables even miss an underlying physical connection. For example, how should T and S at the northern boundary (65°N) influence the NWS? Why isn't the western boundary taken into account instead, which is much more relevant for water mass properties in the eastern North Atlantic.

Being aware of the many studies investigating the influence of NAO variations on the physical oceanic and atmospheric conditions of the NWS, the only thing I have learned from reading the manuscript is that the predictive skill for NWS T and S based on a forecast GCM (GloSea5) NAO index is very low. Other presented conclusions are either trivial or speculative.
I conclude that the study does not provide significant scientific advances, innovations or insights and therefore I do not see a sufficient novelty worth for publication in Ocean Science. The authors mention that they plan to conduct a dynamical downscaling of a GCM forecast. This indeed would be a very interesting experiment for directly assessing forecast skills of the dynamical downscaling approach.

---

## Referee Comment (RC2) · Anonymous Referee #2 · 26 Feb 2018

General remarks: The paper investigates different possibilities to perform seasonal forecasts for the Northwest European shelf. Three methods are explored: a) a direct employment of the global model output, b) an empirical downscaling using large-scale parameters, c) a dynamical downscaling making use of a regional model.

The paper is written in a clear and concise manner. Although it is not extremely innovative, it contains however, several new aspects, which for my opinion deserve publication.

I have just one general criticism, which should be considered seriously by the authors. The data which have been employed for this study just cover the English Channel and

south-western North Sea. Hence, the entire effect of the inflow through the north-western entrances to the North Sea is neglected. Since this north-western inflow contributes around 90% of the total inflow into the North Sea, the major component of the interaction between the Northwest Atlantic and the North Sea is ignored by this means. Therefore, I would strongly propose to include other available long-term data sets from other stations in the analysis, e.g. the time series from Helgoland Roads, Ferry Box data from other routes, etc.

The rest of the comments listed below just concern several minor details:

Detailed Comments: Page 5, line 11: I would assume that the SST assimilation in the CO5 reanalysis also disturbs the heat fluxes. Please give some information about this problem. Page 6, line 16: Please complete the sentence. Page 6, lines 20 to 24: Please discuss the implications of both mentioned problems regarding the riverine forcing. Page 9, line 19: Obviously, due to the he missing tides not enough turbulence is introduced into the system. This should be stated here. Page 9, line 27: As already mentioned above, here, I see the main weakness of the paper. By this approach only processes related to the much less important southern entrance of the North Sea can be considered in detail. As recommended above, the authors should try also to look at other locations, also in the northern North Sea. Page 10, line 28: Sentence unclear. Page 11, line 15: Probably another reason for this behaviour is the fact that the different catchment areas are located in slightly different climate regimes, which also leads to an unclear signal of the run-off. Page 11, line 33: The SSS has not discussed before. Therefore its first mentioning at this place is a little surprising. Page 12, lines 26-27. This sentence needs to be substantiated. Otherwise it is just a platitude. Page 12, line 33: Per se, it is not clear that boundary conditions are most important for an improved dynamical downscaling. Please give some explains. Also the effect of better represented regional atmospheric features is definitely of importance. Page 13, line 4: A repetition of the previous comment. In particular, if an uncoupled dynamical downscaling is performed, also the regional atmospheric resolution will be of importance.

Page 13, line 27: Here, it should be mentioned that the impact of the major inflow is not considered appropriately, due to the choice of stations which are analysed. Page 13, line 28 and 29: This statement is misleading. As I understood the employed models include the influence from the Baltic Sea. Page 13, line 33: The exchange with the Atlantic has already been discussed before. Therefore, it is confusing to say, it will be considered later. Page 14, line 5: Such mentioned advective process from the open boundaries would imply that a considerable time-lag between the boundary signal and the reaction pattern in the North Sea exists. This should be discussed. Page 15, line 10: This argument is not convincing. In principle, also opposing effects can be formulated empirically. Page 15, line 16: To my mind also coupled atmosphere-ocean models could lead to a significant improvement, since regional atmospheric features and the ocean-atmosphere interaction is resolved more realistically in these types of models. Page 16, line 14: This statement is formulated very negative, and, it does not really reflect the general content of entire paper. Therefore, I would suggest to weaken it a bit.

---

## Author Comment (AC1) · 13 Mar 2018

Dear Referee 1,

Thank you for your review, and sorry for the slow response. I wanted to post a reply before making the changes and resubmitting the paper.

The main aim of the paper was to provide a roadmap for North West European shelf seas seasonal forecasting, with a marine management, marine policy and the marine user etc. audience in mind, as well as a general science readership. With this in mind, we wanted to take the reader through the various approaches in a clear manner. We

will make this aim of the paper clearer in the manuscript. We agree that some of the results are not new, but we feel that the description of the approach to NWS seasonal forecasting is novel and should be published within the literature

While the deficiencies of global ocean models are simulating regional seas are well known to the scientific audience, we wanted to clarify that direct use of the seasonal forecasting system is not possible, as there has been a lot of interest in using North Sea seasonal forecasts from marine users.

We will improve our literature study of the paper, and should of referred to more NAO shelf seas response papers. Did you have any papers in particular in mind?

As the English Channel SST forecast was simply demonstrative, we didn't include the equation, but we can do.

We state that dynamic downscaling as a third approach to seasonal forecasting for the NW European shelf seas. Although previous studies state relationships between the open ocean and the NW European shelf seas, it is important to show that these relationships exist within this model and it boundary conditions. The examples chosen for the ocean lateral boundary condition relationship were indicative of the north south difference, rather than to discuss the dominant mechanisms, which was not the purpose of the paper. We removed additional description and discussion of the relationships between the ocean boundaries and the shelf seas response, as it didn't add to the story of the paper.

Thank you again for your review

Kind Regards

Jonathan Tinker et al.
* * *

---

## Author Comment (AC2) · 13 Mar 2018

Dear Referee 2,

Thank you for your review. I wanted to post a reply before making the changes and resubmitting the manuscript.

You raise in important criticism about the data used to study the southern North Sea and the English Channel, and the absence of data influenced by the northern inflow of the North Sea. I understood your point as to be in relation to the reanalysis evaluation section, and think that you raise a very good point. I again point out that the

extensive reanalysis evaluation is undertaken by Copernicus, but will investigate other time-series to compare too. Thank you for the recommendation of the Helgoland Road time series.

You have many other good points which we will address in the resubmitted manuscript

Kind Regards

Jonathan Tinker et al.

---

## Author Comment (AC3) · 26 Mar 2018

Dear all,

We believe that a revised paper can address the reviewers' concerns. In particular:

We will make it clearer that Section 3.1 demonstrates the impact of known deficiencies in global seasonal forecast systems for shelf seas predictions. While we agree with the reviewer that this will come as no surprise to experienced shelf seas scientists, the point may be less well recognised in the global modelling community. In any case we provide an explicit demonstration of the extent of this limitation in a state of the art

global seasonal forecast system. [Reviewer 1]

We will strengthen the discussion of physical mechanisms for some of the empirical relationships shown (we removed some material on this from our original submission in the interests of brevity) [Reviewer 1]

We will include evaluation of the reanalysis against further observed timeseries, as suggested by Reviewer 2.

Kind Regards

Jonathan Tinker et al.

---

## Author Response (AR1)

02/05/2018

Dear Dr. Hoppema,

5  We would like to thank the reviewer for their thoughtful reviews. There contributions have strengthened out paper. We have responded to each of there points below, which are reflected in the manuscript. We have also included a copy of the manuscript which should include the relevant changes highlighted with track changes.

Kind Regards

Jonathan Tinker *et al.*

**Reviewer 1**

Our aim in this paper is to draw evidence on the prospects of seasonal forecasting from the literature and from our own
15  research. We recognise that some aspects will not come as a surprise to some readers; nonetheless, we believe that an overall perspective hasn't been presented before.

This may not have been clear in the original manuscript, given the spirit of the comments of Reviewer 1, and so we have added this into the section "scope of the study" to highlight his point to the reader.

20  We hope that this clarifies the purpose of the paper in response to Reviewer 1's general comments.

We now address the comments of Reviewer 1, paragraph by paragraph (after their introduction paragraph).

**Point 1**

*The first approach, that is using a global circulation model (GCM), is rejected by the authors by referring to model*
25  *limitations of GCMs. However, this rejection is trivial. Since the 1980s the development of regional circulation models (RCM) is motivated by limited representations of shelf sea dynamics in GCMs. Moreover the authors only highlight the deficiency of GCMs in simulating tidal mixing due to the neglect of tidal waves, which is also trivial and known.*

**Author's response:**

While the deficiencies of global ocean models are simulating regional seas are well known to the scientific audience, we
30  wanted to clarify that this means that the direct use of the seasonal forecasting system is not possible – there has been a lot of interest in using North Sea seasonal forecasts from marine users who may not have detailed understanding of the limitations of ocean modelling. While the rejection of the direct approach due to the global model's representation of NWS stratification is simple, the tidal mixing and stratification regimes of the NWS are dominate features of the NWS hydrography and are very important for the ecosystem of the NWS, and the potential end users.

35  **Point 2**

*Their account on the second approach, that is developing empirical relationships between predictable large-scale driving mechanisms and NWS conditions, is a mere correlation analysis of the NAO with reanalysis atmospheric forcing fields and T and S for various sectors of the NWS and eastern North Atlantic. Moreover, the found correlations are all well-known but in*

*the manuscript often not even cited. It is mentioned that correlations do not guarantee causal relations. However, underlying physical mechanisms linking the presented variables are poorly discussed.*

**Author's response:**

The correlations between the driving data and the NWS have been sufficient to make our point. More complex analysis is beyond the scope of the paper.

We have now referenced relevant papers.

We have discussed the mechanisms enough to show that the relationships are plausible; however, as this is not the main focus of the paper, a detailed discussion is not included.

**Point 3**

*Their exemplified "forecast" of English Channel SST based on the NAO index is only a comparison of SST and NAO index time series, quantified by the correlation coefficient. A suggestion how to construct SST from a predicted NAO index, though, is not presented.*

**Author's response:**

As we wanted to show how these relationships may be the basis of a very simple statistical forecast, the relationship between the SST and NAO was sufficient to elucidate our point. However, it's a good point about the equations - we have now put them in the figure.

**Point 4**

*The third approach, that is a dynamical downscaling of a GCM forecast by a RCM, is not yet presented but theoretically discussed by correlation analysis of annual mean T and S on the shelf with various boundary forcings for a regional reanalysis product. The idea behind this is that if the boundary forcings are significantly correlated with the NWS sectors and there is forecast skill by the GCM for large-scale driving mechanisms then a dynamical downscaling should be able to improve the forecast for the NWS. However, just as for the second approach, the found correlations are not new and sometimes the correlated variables even miss an underlying physical connection. For example, how should T and S at the northern boundary (65°N) influence the NWS? Why isn't the western boundary taken into account instead, which is much more relevant for water mass properties in the eastern North Atlantic.*

**Author's response:**

We again note the opening comment that this paper aims to give an overview of the prospects for seasonal forecasts, with evidence from the literature and our research.

We therefore outline the third approach, dynamic downscaling, and note that the NWS must respond to the boundary conditions for such an approach to be possible.

Although previous studies state relationships between the open ocean and the NW European shelf seas, it is important to show that these relationships exist within this model and it boundary conditions.

The example chosen for figure 11 was to highlight the difference in the response of the northern and southern regions to the atmospheric forcings. As we have tended to focus on the atmospheric forcings, we had simply chosen a northern and southern oceanic lateral boundary region for consistency. In response to your comment on the western boundary, we have replaced the northern boundary in figure 11.

**Point 5**

*Being aware of the many studies investigating the influence of NAO variations on the physical oceanic and atmospheric conditions of the NWS, the only thing I have learned from reading the manuscript is that the predictive skill for NWS T and S based on a forecast GCM (GloSea5) NAO index is very low. Other presented conclusions are either trivial or speculative.*

**Author's response:**

This is again relates to our need to clarify the scope of the paper, and our aim being to outline the prospects of NWS seasonal forecasting.
We hope that the new text in the "scope of the study" will clarify our aims for the reader.

**Reviewer 2**

Thank you for your review.
You make a very good point with your general criticism, and in response we have undertaken significant additional model evaluation. Thank you for your suggestions, as we feel this strengthens the paper considerably.
We have responded to your other comments, point-by-point below.

**Page 5, line 11:**

*I would assume that the SST assimilation in the CO5 reanalysis also disturbs the heat fluxes. Please give some information about this problem.*

**Author's response:**

The SST assimilation will improve the SST field, and will affect (improve) the upward radiation. The downward radiation that we consider in this paper is not affected.
I have said this in the text.

**Page 6, line 16:**

*Please complete the sentence.*

**Author's response:**

Rewritten

**Page 6, lines 20 to 24:**

*Please discuss the implications of both mentioned problems regarding the riverine forcing.*

**Author's response:**

Rewritten

**Page 9, line 19:**

*Obviously, due to the missing tides not enough turbulence is introduced into the system. This should be stated here.*

**Author's response:**

I have added this to section 2.3, and added a comment here.

**Page 9, line 27:**

*As already mentioned above, here, I see the main weakness of the paper. By this approach only processes related to the much less important southern entrance of the North Sea can be considered in detail. As recommended above, the authors should try also to look at other locations, also in the northern North Sea.*

**Author's response:**

This is a very good point, thanks for suggesting it.
We have extended the evaluation section significantly to include some mode northern stations.

**Page 10, line 28:**

*Sentence unclear.*

**Author's response:**

Hopefully clarified

**Page 11, line 15:**

*Probably another reason for this behaviour is the fact that the different catchment areas are located in slightly different climate regimes, which also leads to an unclear signal of the run-off.*

**Author's response:**

If I have understood your point, it's a good one – I have added it earlier in the text.

**Page 11, line 33:**

*The SSS has not discussed before. Therefore its first mentioning at this place is a little surprising.*

**Author's response:**

Sea surface salinity is briefly discussed at the end of the previous paragraph. I have expanded on this a little.
If you mean the acronym SSS rather than the variable sea-surface salinity, I have redefined it in the previous paragraph.

**Page 12, lines 26-27.**

*This sentence needs to be substantiated. Otherwise it is just a platitude.*

**Author's response:**

This has been rewritten.

**Page 12, line 33:**

*Per se, it is not clear that boundary conditions are most important for an improved dynamical downscaling. Please give some explains. Also the effect of better represented regional atmospheric features is definitely of importance.*

**Author's response:**

They are not the most important thing for a successful NWS seasonal forecast based on dynamically downscaling a seasonal-forecast, the seasonal forecast is. However, it's important that the NWS will respond to the large-scale external environment, in which the seasonal forecast system does have skill – if it is very chaotic, and an initial conditions problem, we would not
5   necessarily expect the NWS to response to any predicted variability from the seasonal forecasting system. I have hopefully clarified this in the text.

**Page 13, line 4:**

*A repetition of the previous comment. In particular, if an uncoupled dynamical down-scaling is performed, also the regional atmospheric resolution will be of importance.*

10  **Author's response:**

See the response to the previous comment.

**Page 13, line 27:**

*Here, it should be mentioned that the impact of the major inflow is not considered appropriately, due to the choice of stations which are analysed.*

15  **Author's response:**

Hopefully this is not necessary with the additional reanalysis evaluation.

**Page 13, line 28 and 29:**

*This statement is misleading. As I understood the employed models include the influence from the Baltic Sea.*

**Author's response:**

20  Good point, this has been removed

**Page 13, line 33:**

*The exchange with the Atlantic has already been discussed before. Therefore, it is confusing to say, it will be considered later.*

**Author's response:**

25  Happy to remove the phrase saying we'll discuss it later.

**Page 14, line 5:**

*Such mentioned advective process from the open boundaries would imply that a considerable time-lag between the boundary signal and the reaction pattern in the North Sea exists. This should be discussed.*

**Author's response:**

30  We have done additional analysis for this, with lag correlation, and summarise the findings here. However as this study focuses on the surface fluxes, consideration of these lags doesn't impact the story of the paper much.

**Page 15, line 10:**

*This argument is not convincing. In principle, also opposing effects can be formulated empirically.*

**Author's response:**

That's true.

I have amended the text.

**Page 15, line 16:**

*To my mind also coupled atmosphere-ocean models could lead to a significant improvement, since regional atmospheric features and the ocean-atmosphere interaction is resolved more realistically in these types of models.*

**Author's response:**

We agree that this could be important, and have a model in development that may allow this to be considered. However, we think that assessment of this model is beyond the scope of this paper.

We have altered the text, say that this is a key area, rather than the key area.

**Page 16, line 14:**

*This statement is formulated very negative, and, it does not really reflect the general content of entire paper. Therefore, I would suggest to weaken it a bit.*

**Author's response:**

Thanks, I have weakened it.

**What are the prospects for seasonal prediction of the marine environment of the Northwest European shelf?**

**Abstract.**

5 Sustainable management and utilisation of the Northwest European Shelf Seas (NWS) could benefit from reliable forecasts of the marine environment on monthly-to-seasonal timescales. Recent advances in global seasonal forecast systems, and regional marine reanalyses for the NWS, allow us to investigate the potential for seasonal forecasts of the state of the NWS. We identify three possible approaches to address this issue: A) basing NWS seasonal forecasts directly on output from the Met Office's GloSea5 global seasonal forecast system; B) developing empirical downscaling relationships between large-
10 scale climate drivers predicted by GloSea5, and the state of the NWS; and C) dynamically downscaling GloSea5 using a regional model. We reject A) after showing that the GloSea5 system is inadequate for simulating the NWS directly. Turning to B), we explore empirical relationships between the winter North Atlantic Oscillation (NAO), and NWS variables estimated using a regional reanalysis. We find some statistically significant relationships, and present a skilful prototype seasonal forecast for English Channel sea surface temperature.

15 We then consider the potential of C). We find large scale relationships between inter-annual variability in the boundary conditions and inter-annual variability modelled on the shelf, suggesting that dynamic downscaling may be possible. We also show that for some variables there are opposing mechanisms correlated to the NAO, for which dynamic downscaling may improve on the skill possible with empirical forecasts. We conclude that there is potential for the development of reliable seasonal forecasts for the NWS, and consider the research priorities for their development.

20 **Copyright Statement**

The works published in this journal are distributed under the Creative Commons Attribution 4.0 License. This licence does not affect the Crown copyright work, which is re-usable under the Open Government Licence (OGL). The Creative Commons Attribution 4.0 License and the OGL are interoperable and do not conflict with, reduce or limit each other.

© Crown copyright 2018

[revised manuscript text omitted]

**Comment [MOU1]:** Review 2:

**Page 5, line 11:**
*I would assume that the SST assimilation in the CO5 reanalysis also disturbs the heat fluxes. Please give some information about this problem.*
**AUTHORS RESPONSE:**
The SST assimilation will improve the SST field, and will affect (improve) the upward radiation. The downward radiation that we consider in this paper is not affected. I have said this in the text.

[revised manuscript text omitted]

**Comment [MOU2]:** Reviewer 2:

**Page 6, line 16:**
*Please complete the sentence.*
**AUTHORS RESPONSE:**
Rewritten.

[revised manuscript text omitted]

**Comment [MOU3]:** Reviewer 1:

**Point 1**
*The first approach, that is using a global circulation model (GCM), is rejected by the authors by referring to model limitations of GCMs. However, this rejection is trivial. Since the 1980s the development of regional circulation models (RCM) is motivated by limited representations of shelf sea dynamics in GCMs. Moreover the authors only highlight the deficiency of GCMs in simulating tidal mixing due to the neglect of tidal waves, which is also trivial and known.*
**AUTHORS RESPONSE:**
While the deficiencies of global ocean models are simulating regional seas are well known to the scientific audience, we wanted to clarify that this means that the direct use of the seasonal forecasting system is not possible – there has been a lot of interest in using North Sea seasonal forecasts from marine users who may not have detailed understanding of the limitations of ocean modelling. While the rejection of the direct approach due to the global model's representation of NWS stratification is simple, the tidal mixing and stratification regimes of the NWS are dominate features of the NWS hydrography and are very important for the ecosystem of the NWS, and the potential end users.

**Comment [MOU4]:** Reviewer 2:

**Page 9, line 19:**
*Obviously, due to the missing tides not enough turbulence is introduced into the system. This should be stated here.*
**AUTHORS RESPONSE:**
I have added this to section 2.3, and added a comment here.

Further evidence is shown under Question 2 below that using GloSea5 NWS fields directly would be problematic. We therefore conclude that this approach (Approach A) is not viable, and some form of downscaling of the GloSea5 fields would be needed to generate reliable NWS forecasts.

**3.2. Question 2: How well does the CO5 NWS reanalysis represent inter-annual variability on the NWS?**

While the ability of the CO5 NWS reanalysis to simulate the mean state of the NWS is thoroughly evaluated (O'Dea *et al.* 2012; Wakelin *et al.* 2014), its ability to simulate inter-annual variability has received less attention. Evaluation requires long observed time-series, preferably of variables that are not assimilated into the reanalysis. Here we focus on 5 locations across the NWS: the Harwich to Hook of Holland ferry box in the southern North Sea; the Western Channel Observatory (WCO) in the English Channel; the Malin Head weather station, north of Ireland; the Fair Isles time-series (northeast of Scotland), and then Helgoland Roads time-series in the southern North Sea. All these datasets are described in the Methods section.

First, we compare the observed time-series of surface salinity in the southern North Sea Ferry Box to that from GloSea5 and the CO5 NWS reanalysis (Figure 3). The time-series exhibits multi-year oscillations and these are well simulated by the NWS Reanalysis (r = 0.89, p = 0.00), despite the fact that it does not assimilate salinity observations. There is a fresh bias in the model (-0.20) and a slightly greater variation (standard deviation ratio of 1.11). GloSea5 does not capture a realistic multi annual variability (r = 0.2, p = 0.00 with standard deviation ratio of 1.81) and modelled salinity also shows a large fresh bias that increases due to a substantial salinity drift over the duration of the time-series – further evidence that direct reading from NWS fields from GloSea5 would be problematic. As there are differences in the river forcings between GloSea5 and the CO5 NWS reanalysis we would expect differences in the modelled salinity. The CO5 NWS reanalysis uses E-HYPE river forcing (Donnelly *et al.* 2013) which are specified daily whereas GloSea5 uses a river climatology (Dai and Trenberth 2002; Bourdalle-Badie and Treguier 2006), and so exhibits no inter-annual variability.

[revised manuscript text omitted]

We now investigate empirical forecasts based on the response of the CMEMS reanalysis to the observed NAO, and then applied to the GloSea5 forecast NAO.

**3.3. Question 3: Can predictable climate indices provide real predictive skill for the NWS?**

In the literature, there are many empirical relationships between climate indices and various physical and biological responses. The CMEMS reanalysis (through data assimilation) combines observations with models to give the best possible state estimate of the NWS, and so provides a powerful tool to develop such relationships. We focus on the winter NAO, as it is an important source of year-to-year variability in the NWS, and GloSea5 has predictive skill for the winter NAO. By investigating relationships between the CMEMS reanalysis fields and the observed (NOAA) NAO, and then considering how these relationships change when we use the GloSea5 forecast NAO, we can explore the empirical approach to NWS seasonal forecasting. We note that many of the relationship we find between the NAO and the NWS are not new, and are underlain by published relationships (e.g. Becker and Pauly 1996; Dippner 1997; Hurrell and Van Loon 1997; Hurrell and Deser 2009).

First we focus on shelf temperature, and restrict our analysis to the surface forcing that we consider important for shelf temperatures. We investigate the correlations (and lagged correlations) between the winter (DJF) NAO and this subset of surface forcings surface forcing (Figure 6). We find a positive correlation of the NAO with the DJF surface air temperature (consistent with Hurrell and Van Loon 1997) and humidity (Figure 6a, e), and this persists for one month (to January-March) in most regions (Figure 6b, f). There is a significant negative correlation between DJF NAO and the incoming solar radiation for shelf regions west of the UK (Figure 6m), although this only persists to a significant level in the Irish Shelf (Figure 6n). DJF NAO is strongly positively correlated to the downward component of thermal radiation for most shelf regions west of the UK and for the Norwegian Trench (NT), and this persists for 3 months in some regions (English Channel) (Figure 6i-l).

We now consider the surface forcings we think are likely to be important for shelf salinity (Figure 7). The DJF NAO is strongly correlated with the DJF 10 m wind magnitude across the domain (consistent with Hurrell and Deser 2009), and this persists into the third month (March-May) for the southern and central North Sea (Figure 7i-l). The correlation between winter (DJF) NAO acts in opposite ways for winter (DJF) mean sea level and for total precipitation. The DJF mean sea-level pressure (total precipitation) is negatively (positively) correlated with DJF NAO in the northern regions and positively (negatively) correlated in the southern regions. These correlations persist for a few months in some regions (Figure 7a-d, e-h), and are consistent with the correlations of Hurrell and Deser (2009).

River systems can give additional predictability by continuing to respond after the forcing, or can reduce predictability by having such long response times that they act as a low-pass filter. Furthermore, different river catchment areas are located in

**Comment [MOU6]:** Reviewer 1: Point 2
*Their account on the second approach, that is developing empirical relationships between predictable large-scale driving mechanisms and NWS conditions, is a mere correlation analysis of the NAO with reanalysis atmospheric forcing fields and T and S for various sectors of the NWS and eastern North Atlantic. Moreover, the found correlations are all well-known but in the manuscript often not even cited. It is mentioned that correlations do not guarantee causal relations. However, underlying physical mechanisms linking the presented variables are poorly discussed.*
**AUTHORS RESPONSE**
The correlations between the driving data and the NWS have been sufficient to make our point. More complex analysis is beyond the scope of the paper.
We have now referenced relevant papers. We have discussed the mechanisms enough to show that the relationships are plausible; however, as this is not the main focus of the paper, a detailed discussion is not included.

different climate regimes, and so can respond in different ways, which can further complicate the response. The river runoff forcings in the Norwegian Trench (Figure 7m-p) are highly correlated with the DJF NAO, and this persists until the following summer (July-September, not shown). The runoff in regions influenced by northern British and Irish rivers (e.g. Irish Shelf) is also positively correlated with the NAO and shows persistence beyond the winter season. The regions including much of the European coast (Armorican Shelf, English Channel, and southern North Sea) show little correlation of runoff with the NAO, perhaps reflecting the larger catchments not having time to respond to the NAO, and their more southerly location. The river correlation patterns were consistent with those of Rödel (2006) and Bouwer *et al.* (2008).

Having shown the correlations of the NAO with the important surface forcings, we now look directly at the relationship between the observed DJF NAO and the shelf response (Figure 8). We find a significant positive correlation between the winter NAO and the winter SST in the most southern and eastern shelf regions (Figure 8a), consistent with previous studies (Dippner 1997; Hurrell and Van Loon 1997). In most of these regions, the significance of these correlation persists for one month (Figure 8b), and in the English Channel and southern North Sea a second month (February-April (FMA) SST, Figure 8c). The NBT correlations also show significant correlations with the NAO and having memory in some regions. The DJF NAO is generally not significantly correlated with SSS (Sea-Surface Salinity) in most regions, although there is a significant correlation in the Skagerrak/Kattegat, which persists until FMA (Figure 8e-h). There could be a much lower frequency salinity response to the NAO (e.g. Mysak *et al.* 1996; Belkin *et al.* 1998), which would not be captured by these correlations. Such a response may provide predictability on longer time scales.

The above results suggest that knowledge of the NAO index could provide some skill for important variables, at modest lead times of 1-2 months, even if the DJF NAO is only determined from observations (at the end of the December-February period). Because the GloSea5 system has skill in predicting the DJF NAO index from the previous November (Scaife *et al.* 2014), it is possible that the lead time could be increased by using the predicted rather than the observed NAO index. In Figure 9 we examine correlations for the same predicted NWS variables as in Figure 8, but this time using the DJF NAO index predicted from the ensemble mean of the GloSea5 forecast run the preceding November. Unsurprisingly the correlations are generally lower than for the observed NAO index, and many are not statistically significant at the 95 % level. However, the correlations are largely of the same sign and pattern as for the observed NAO index, and remain significant. This suggests that a prototype seasonal forecast based on the GloSea5 NAO may be possible. For SST and NBT, there are regions with exhibit persisted significant correlations (e.g. English Channel) which is promising. The correlation patterns for SSS are however, quite different to those from the observed NAO. There is a general negative correlation across the shelf that gets stronger with an increasing lag. The persistence in the NAO-SSS correlation reflects the longer term nature of salinity anomalies. The difference in the SSS correlation patterns between the observed and GloSea5 NAO perhaps act as an error estimate to this approach, suggesting caution and further assessment is needed before relying on an empirical seasonal forecast of this form for SSS. Overall, the results with the GloSea5 NAO suggest that real relationships exist between the forecast NAO and the observed NWS fields, and that further improvements in the seasonal NAO forecast would deliver higher levels of forecast skill and/or regional detail.

The correlations between the NAO and the shelf response describe how strong a linear relationship exists between the two. Where there is significant skill (a significant correlation) this linear relationship can be used to predict shelf response from the NAO. Here we show an example for one of the stronger NAO/shelf response correlations: English Channel SST. We have forecast the temperature based on both the observed NAO and GloSea5 forecast, and plotted this against the modelled temperature in Figure 9 (both have been normalised) with zero lag and the following 4 monthly lagged seasons (forecasting JFM, FMA etc. based on the DJF NAO). We also include the time series time-series of the observed and GloSea5 DJF NAO for comparison. This figure shows how an NAO based seasonal forecast based on the relationships discussed in this paper would look.

**Comment [MOU7]:** Reviewer 2:
**Page 11, line 15:**
*Probably another reason for this behaviour is the fact that the different catchment areas are located in slightly different climate regimes, which also leads to an unclear signal of the run-off.*
**AUTHORS RESPONSE:**
If I have understood your point, it's a good one – I have added it earlier in the text.

**Comment [MOU8]:** Reviewer 2:
**Page 11, line 33:**
*The SSS has not discussed before. Therefore its first mentioning at this place is a little surprising.*
**AUTHORS RESPONSE:**
Sea surface salinity is briefly discussed at the end of the previous paragraph. I have expanded on this a little.
If you mean the acronym SSS rather than the variable sea-surface salinity, I have redefined it in the previous paragraph.

[revised manuscript text omitted]

**Comment [MOU18]:** Reviewer 2:
**Page 15, line 10:**
*This argument is not convincing. In principle, also opposing effects can be formulated empirically.*
**AUTHORS RESPONSE:**
That's true.
I have amended the text.

**Comment [MOU19]:** Reviewer 2:
**Page 15, line 16:**
*To my mind also coupled atmosphere-ocean models could lead to a significant improvement, since regional atmospheric features and the ocean-atmosphere interaction is resolved more realistically in these types of models.*
**AUTHORS RESPONSE:**
We agree that this could be important, and have a model in development that may allow this to be considered. However, we think that assessment of this model is beyond the scope of this paper.

We have altered the text, say that this is a key area, rather than the key area.

[revised manuscript text omitted]

---

## Author Response (AR2)

Dear Mario,

Thank you for your additional suggestions. We hope we have clarified manuscript, and addressed your concerns.

We have re-plotted most of the figures, and believe they are much clearer now than when we last submitted. We hope you are happy with them now.

We have added additional paragraphs to address some of your points, and altered the text/figures in response to other points.

All our changes should be clear with track changes.

We look forward to hearing from you and please contact us if you have any other queries,

Kind Regards

Jonathan et al.

**Editor (18/06/2018):**

You are using different data sets from other sources. Please check again the fair data use statements of those data providers whether the acknowledgements are sufficient and correct.

**Response (05/07/2018)**

I have contacted the ICES data providers to confirm that they are happy with my updated citations.

**Editor (18/06/2018):**

The references are not according to the Ocean Science format. Some are incomplete, for example:

Bourdalle-Badie, R. and Treguier, A. M. (2006). Please go through the references for possible errors or incompleteness.

**Response (05/07/2018)**

I have changed the reference format, and checked for incomplete references

**Editor (18/06/2018):**

Referee #1 point 1: What did you change in the manuscript following this comment?

**Response (27/06/2018)**

In this paper we are trying to outline different approaches for seasonal forecasting on the NWS. The first approach, the direct use of GloSea5 data, should not be mindless used, as its global ocean has a poor representation of the NWS. Reviewer 1 is correct that our dismissal is trivial, but we think this is a clear way to illustrate GloSea5's NWS limitations to our wider audience.

The absolute rejection this approach is not a key point of our paper, so having made our point on its limitations, we have now softened the text on the manuscript on this, and even suggest additional research is required to show when this approach may be valid.

This point also highlighted that we had not managed our readers' expectations of this paper. We have therefore also added the following paragraph to "1.3 Scope of the study"

> "Our aim in this paper is to draw evidence on the prospects of seasonal forecasting from the literature and from our own research. We recognise that some aspects will not come as a surprise to some readers, nonetheless, we believe that an overall perspective hasn't been presented before."

**Editor (18/06/2018):**

Referee #1 point 2: I agree with the authors that a more complex analysis is not needed here. However, one could also briefly discuss what kind of physical mechanisms are at play. I encourage the authors to expand the discussion in this respect to some extent.

**Response (2/7/2018)**

This is a good point. On re-reading, I realise that we describe the patterns of the correlation in section 3.3, without any explanation of likely mechanism. I have added sentences throughout this section relating the correlation patterns to the regional implications of NAO (i.e. drier winters vs. wetter winters).

**Editor (18/06/2018):**

Referee #1 point 3: I think this is a legitimate argument by the referee. It would be good if you could go deeper into this.

**Response (2/7/2018)**

In this section we wanted to show how a simple empirical forecast based on the NAO would work. We have now added a paragraph explaining how to create such an empirical forecast.

Reviewer 1 had a point that we were just looking at the two time series as we were defining different equations for the observed and forecast NAO. We have now calculated the equations for the observed NAO and applied them to the forecast NAO.

We have replaced the previous paragraph with the following two:

> "The correlations between the NAO and the NWS fields describe how strong a linear relationship exists between the two. Where there is significant skill (a significant correlation, Figure 8) this linear relationship can be used to predict the NWS fields from the NAO. A simple approach would be to find the slope and intercept between the observed (NOAA) NAO and the shelf seas variable, and then apply this equation to the GloSea5 forecast NAO. This provides a simple empirical forecast giving information about the future state of the NWS (e.g. greater than average, less than average). Other non-linear relationships (e.g. quadratic etc.) may exist between the NAO and NWS fields that could be used as the basis of an empirical forecast – further analysis (and curve-fitting) would be required for their identification. The reanalysis provides a coherent dataset in order to explore such relationships.
> Here we present such an example forecast of the English Channel SST initialised in November, for the following winter. Using the reanalysis, we construct a linear empirical relationship between the observed (NOAA) DJF NAO and the English Channel winter (DJF) SST. The correlation in Figure 8 quantifies the strength of this linear relationship, and its significance. We then apply this equation to the GloSea5 forecast NAO (Figure 10). Due to the persistence of the NAO SST correlation in this region, we are able repeat the process to extend the lead time (February-April forecast initialised in November), beyond which the underlying correlations significance is greater than 0.05. These (normalised) forecasts are illustrated in Figure 10."

**Editor (18/06/2018):**

Referee #1, point 4: Please answer this question by the referee: "However, just as for the second approach, the found correlations are not new and sometimes the correlated variables even miss an underlying physical connection. For example, how should T and S at the northern boundary (65°N) influence the NWS?" This is an important point.

**Response (27/06/2018)**

We have taken out the Northern Boundary (correlations with this area might have been due to outflow of the if the NWS via the Norwegian Trench causing the T and S at the northern boundaries, or 2 things correlated with a third thing). We now include the western boundary (as the reviewer requested) and discuss how lateral oceanic boundary conditions can affect the NWS conditions. We have already discussed how the boundary salinity affects the salinities on the shelf, but I have now added text to clarify how the lateral temperature boundaries affect the temperatures on the shelf:

> "Water advected from the lateral boundaries transports heat and salt onto the shelf, which influences these correlations. Heat is exchanged with the atmosphere faster than freshwater, and the memory of temperature of the lateral boundaries is overwhelmed by the surface heat exchange, while salinity memory persists longer. Therefore, the shelf SST correlations with the oceanic temperature boundary conditions is likely to be due to common surface forcings acting on the open ocean and shelf seas temperatures"

**Editor (18/06/2018):**

Referee #2 Page 5, line 11:

"I would assume that the SST assimilation in the CO5 reanalysis also disturbs the heat fluxes. Please give some information about this problem.

Author's response: The SST assimilation will improve the SST field, and will affect (improve) the upward radiation. The downward radiation that we consider in this paper is not affected. I have said this in the text."

Editor: I am not convinced by your reply. Please explain better, especially in the manuscript.

**Response (27/06/2018)**

The SST assimilation will affect the upward and therefore the net radiation fluxes. We do not consider either the upward or net fluxes in this study, only the downward component of the radiation fluxes. These are prescribed from the ERA atmospheric reanalysis and therefore the SST assimilation will not affect our correlations. If the reanalysis was a coupled system, the SST and so the net radiation fluxes could feed back on to the atmosphere, but this is not possible in a uncoupled (one-way forced) ocean-only reanalysis.

I have not clarified this in the text in section 2.1:

> "Furthermore, the CORE bulk formulae receive prescribed downward component of the short-wave- and long-wave- radiation, and calculate the upward component internally using modelled SST. Therefore, while SST data assimilation will improve the SST, and affect the upward radiation, the downward radiation is not affected by the SST assimilation. In sections 3.3 and 3.4 of this study, we consider correlations of shelf sea variables with the prescribed *downward* component of the radiative fluxes only."

And have added text into section 3.3, where I am discussing the radiative fluxes. The added text is:

> "Note that the SST assimilation increments will affect the upward radiation fluxes (which are calculated from the modelled SST), but, as the reanalysis system is an ocean-only uncoupled system, the downward component of the radiation (which we consider here) is prescribed (by the ERA interim data), and so these correlations are not affected by this assimilation."

**Editor (18/06/2018):**

P9, L27 The sentence does not seem to be correct. Should it start with "Giving"? Please rewrite the sentence.

**Response (2/7/2018)**

I have changed this to:

> "From the answers to (1-4), we conclude by assessing the prospects for seasonal prediction on the NWS and the likely pros and cons of empirical and dynamical downscaling approaches (B) and (C), and suggest near-term priorities for applications and research."

**Editor (18/06/2018):**

P12 ,L30-31 The link between the output and the abbreviations is not clear at all

**Response (18/06/2018):**

Good point, I have now clarified these.

**Editor (18/06/2018):**

P12, last paragraph (and continuing on next page) under 2.5.1 Every sentence starts with "We". Please try to change this to enhance the readability.

**Response (18/06/2018):**

I have rewritten this, removing the 'We's'

**Editor (18/06/2018):**

Section 2.5.1 "We compare time-series of the shelf response to time-series of the drivers to identify important relationships. We note that a statistically significant correlation does not imply a causal relationship, and so we interpret the spatial patterns of the correlation coefficients to help interpret the underlying mechanisms behind the correlations. We consider it beyond the scope of this study to undertake sensitivity studies to explore any mechanisms in detail." I only partly agree with this. I think sensitivity studies are too much for this exercise indeed. But some ideas about the mechanisms would certainly be most welcome, the more so as relationship

only based on correlations might just be showing things that make no sense at all, without any physical background.

**Response (05/07/2018)**

You asked for more description of mechanisms (in response to Reviewer 1's Point 2), to which we added a paragraph in section 3.3.

To reflect that paragraph, and in response to your comment, we have added this to section 2.5.1

> "Some possible mechanisms are described, but it is considered beyond the scope of this study to undertake sensitivity studies to explore any mechanisms in detail."

**Editor (18/06/2018):**

P14, L12 " …of surface salinity in the southern North Sea Ferry Box …" This is strange. I think you mean the salinity of ferry box data.

**Response (18/06/2018):**

I have changed this to:

> "First, we compare the observed time-series of surface salinity in the southern North Sea (from the Ferry Box) to that from GloSea5 and the CO5 NWS reanalysis"

**Editor (18/06/2018):**

P15, L1, L5 Do not use psu because this is not a unit.

**Response (18/06/2018)**

I followed you advice from your email and removed psu from the figure, but, to maintain clarity and readability of the text, retailed them in the text.

**Editor (18/06/2018):**

P19, L1 UV10 may not be common to all readers. Please define.

**Response (18/06/2018)**

I have now clarified this term (see text below), and moved this description earlier in the text (to section 3.3), where wind magnitudes are first discussed:

"(UV10; defined as the magnitude (wind-speed irrespective of direction) of the 10m wind)"

**Editor (18/06/2018):**

P20, L11-12 "Further improvements to the reliability of the NAO forecast that are in principle achievable (e.g. Eade et al. 2014) would be expected to lead to further improvements in the forecasts for NWS regions." This is a little bit vague. The only reason that such a contention could be interesting to the reader is if these improvements are mentioned and explained. Please expand on this to some extent.

**Response (26/06/2018)**

I have expanded on this, to give context.

"Further improvements to the reliability of the GloSea5 NAO forecast may enhance the forecasts for the NWS region. While being skilful, GloSea5 has been shown to be under-confident (meaning there is too high a proportion of unpredictable noise in the forecast) for the NAO and the wider Atlantic region (Eade et al., 2014), and for the inter-annual predictions (Dunstone et al., 2016). This problem is common to most skilful seasonal forecasting systems (Baker et al., 2018), but as there are exceptions, it is not inherent to such systems. Solving the under-confidence issue is an active research area."

**Editor (18/06/2018):**

P20, L13 Is this indeed preliminary? That is not how I understand this paper. You are assessing the possibilities and chose three methods. It would be disappointing if this is only preliminary.

**Response (10/07/2018)**

We have removed the "preliminary".

**Editor (18/06/2018):**

P20, L24 Whole point C. I find the structure to be deviating from the two earlier points. What I expect is a clear assessment whether this method works or not, or only to some extent. Maybe it is contained in the text somehow, but certainly not very clearly. I suggest restructuring this point C.

**Response (10/07/2018)**

On re reading, I agree with you – it was very different from the previous points. I have therefore, rewritten it as:

" C) Dynamic downscaling. For the NWS variables that cannot be skilfully forecast directly from the NAO, additional predictability might be possible by dynamic downscaling. With the dynamic downscaling approach, much more information from the global seasonal forecasting system (including both predictable and unpredictable components) is used. This may allow important subtleties that are not captured in a simple NAO index, to be resolved. Additionally, persistence (encapsulated in the initial conditions of the regional model) can provide additional skill. Our analysis of the relationship between the NWS variability with the boundary drivers corroborates the boundary constrained nature of the NWS (e.g. Holt et al., 2016), and so supports the possibility of additional forecast skill from dynamical downscaling, provided the driving global system can forecast the driving boundary conditions reliably. However, this approach requires significant additional research."

**Figures**

**Editor (18/06/2018):**

Figure 1

The map is not clear at all. It should be bigger, the lines demarcating the coast would better be in a different colour, and the names of some countries should be added for orientation. By the way, what is called "southern North Sea" here is also known as the Southern Bight of the North Sea.

**Response (25/06/2017)**

I have reproduced this figure, and hope it's much clearer now.

Thanks for the information about the German Bight. I have used southern North Sea for consistency with the original Wakelin paper. This region includes both the German bight (the eastern half of this region) and the Southern Bight (the western portion).

**Editor (18/06/2018):**

Figure 3 caption "as 2-year running mean" and not "with 2 year running mean", I think.

**Response (2/7/2018)**

I have changed this.

**Editor (18/06/2018):**

Figure 3 It is not shown if the data are spatially comparable indeed. Which area in CO5 NWS reanalysis and in GloSea5 was used for averaging? And is this exactly the region of the ferry box data?

**Response (25/06/2018)**

These time-series are from the nearest model grid box. I have added this to the caption:

"Both modelled time-series are from the nearest model grid-box to the observations."

and:

"(Figure 3, both from nearest model grid-box)."

to the text

**Editor (18/06/2018):**

Figure 4 caption Twice "observed". Time-series (typo).

**Response (18/06/2018)**

Corrected

**Editor (18/06/2018):**

Figure 5 caption: "Lower right panel shows the location" must be "Upper right panel shows the location"

**Response (18/06/2018)**

Corrected

**Editor (18/06/2018):**

Figure 6 These are really many panels. I am not sure whether this will be readable and understandable at all. If possible, could you split this figure in two or make it more readable in any other way?

Figure 7 Same as for Figure 6

Figure 8 Same as for Figure 6

Figure 9 Same as for Figure 6

**Response (25/06/2018)**

I have reduces them from a 4x4 set of panels to a 3x3 set. Losing the final column wasn't too bad (most of the correlations hadn't persisted that long anyway. For most figures, losing one of the variables (rows) wasn't detrimental to the paper.

**Editor (18/06/2018):**

Figure 10 The panels are too small. It is hardly possible to extract information from these plots. This must be changed.

**Response (25/06/2018)**

I have removed some of the extra lead time panels, and increased the figure width and text size. I think this is much clearer

**Editor (18/06/2018):**

Figure 11 Same as for Figure 6, even more since there are 18 panels in it.

**Response (25/06/2018)**

I have separated these into 3 sub plots – keeping all the original information, but only having 2x3 panels per subplot. I think this is much clearer.

**What are the prospects for seasonal prediction of the marine environment of the Northwest European shelf?**

**Abstract.**

Sustainable management and utilisation of the Northwest European Shelf Seas (NWS) could benefit from reliable forecasts of the marine environment on monthly-to-seasonal timescales. Recent advances in global seasonal forecast systems, and regional marine reanalyses for the NWS, allow us to investigate the potential for seasonal forecasts of the state of the NWS. We identify three possible approaches to address this issue: A) basing NWS seasonal forecasts directly on output from the Met Office's GloSea5 global seasonal forecast system; B) developing empirical downscaling relationships between large-scale climate drivers predicted by GloSea5, and the state of the NWS; and C) dynamically downscaling GloSea5 using a regional model. We show that the GloSea5 system can be inadequate for simulating the NWS directly and so move on from A).  Turning to B), we explore empirical relationships between the winter North Atlantic Oscillation (NAO), and NWS variables estimated using a regional reanalysis. We find some statistically significant relationships, and present a skilful prototype seasonal forecast for English Channel sea surface temperature.

We then consider the potential of C). We find large-scale relationships between inter-annual variability in the boundary conditions and inter-annual variability modelled on the shelf, suggesting that dynamic downscaling may be possible. We also show that for some variables there are opposing mechanisms correlated to the NAO, for which dynamic downscaling may improve on the skill possible with empirical forecasts. We conclude that there is potential for the development of reliable seasonal forecasts for the NWS, and consider the research priorities for their development.

**Copyright Statement**

The works published in this journal are distributed under the Creative Commons Attribution 4.0 License. This licence does not affect the Crown copyright work, which is re-usable under the Open Government Licence (OGL). The Creative Commons Attribution 4.0 License and the OGL are interoperable and do not conflict with, reduce or limit each other.

© Crown copyright 2018

[revised manuscript text omitted]

---

## Author Response (AR3)

Topic Editor Decision: Publish subject to minor revisions (review by editor) (19 Jul 2018) by Mario Hoppema

Comments to the Author:

Dear Jonathan and co-authors,

5 **Editor:**

Your revised manuscript has been much improved and I think it is shortly before publication. I have gone through it and below is my list of final comments, which I encourage you to account for when submitting the final version of your manuscript.

**Response:**

10 Thank you, I feel we now have a much stronger manuscript. I have addressed all your comments, and have used track changes show where they have been addressed in the text.

**Editor:**

For someone who is not really familiar with the region, it is really hard to learn where everything is occurring. In the text

15 some names of coastal towns are mentioned, but also the Scottish and Danish coast, English Channel, Irish Sea, Armorican Shelf, Celtic Sea, for example. It would be most helpful if all these geographical names used would appear in one of the figures. This also holds for the different currents that are mentioned throughout the manuscript.

**Response:**

I have had a very thorough rework of Figure 1, to include the countries names (in letter format), with the observation

20 location. I have also reworked in the lateral boundary regions so they spatially match the map. I think the figure is much improved, and hope you're happier with it!

The figure caption gives all the region names (English Channel, Irish Sea, Armorican Shelf, Celtic Sea).

I don't think I can feasibly add the currents or coastal towns, as the plot would not be readable.

In the text I mention the European slope current and the Fair Isle current. I now describe the location and direction of the

25 current in the text when I first mention them:

"…measure the temperature and salinity (upper 100m) of Atlantic water entering the North Sea between Scotland and the Shetland Islands, via the Fair Isle Current…"

30 "…the tightly defined European slope current (which flows northward, following the continental shelf slope from the south of the domain, through the Bay of Biscay, around Ireland, and then towards Norway, across the top of the North Sea)."

I hope this is sufficient. I could add another figure, simply showing currents, town names, etc, but don't I think this would be

35 a good idea, as there would be so much replication with figure 1.

I have also tidied figure 2, to improve its quality.

**Editor:**

40 P1, L11 We show that the GloSea5 system is inadequate for simulating the NWS directly.

("can be" is strange here) (delete the last part of the sentence: and so move on from A).

**Response:**

Changed

**Editor:**

P1, L15 I suggest: As to point C) we find large-scale …

**Response:**

I have rewritten these few sentences, making it a bit less flowery, and a bit more formal. I have removed:

"… and so move on from A)"

"… Turning to B"

"We then consider the potential of C"

**Editor:**

P5, L12 This is the first time SSS is used. They should be defined here.

**Response:**

Good point, I thought I had defined it on my first use. I have now defined this here.

**Editor:**

P6, L12 Wegener (typo)

**Response:**

Thanks, changed now.

**Editor:**

P6, §2.4.2 and 2.4.3 You are using data from particular locations on the NWS. Some of the time series stations are shown in Fig. 5, but other data (for example, Hoek van Holland-Harwich) are not. Please show them in one of the figures.

**Response:**

I have included this in figure 1 now. In the text I refer the reader to the figure, when introducing the observations data sets.

**Editor:**

P11, L40 Please be more specific with "its skill, resolution etc." What does the etc mean?

**Response:**

I am now more specific, and have changed this to:

The underlying skill of the global seasonal forecasting system would be the basis to any such NWS downscaling, and its skill at predicting the surface and lateral boundary conditions of the NWS will be the leading order limit to the subsequent skill of the NWS seasonal forecast.

**Editor:**

P12, L1 Suggestion: We start to explore … (delete: can; you are going to do that, right?)

**Response:**

Good point, I have changed that now.

**Editor:**

P14, L10-12 I am not sure what this sentence wants to convey.

**Response:**

I have simplified this now.

**Editor:**

P14, L19-26 Many times the word "may" is used here. I think in many cases this is unnecessary and tends to give the reader the impression that nothing is certain or even doubtful. Similar for some cases in the Conclusion section.

**Response:**

Good point. There were three "may"'s. I have removed the first two.

**Editor:**

P14, L28 Here it is stated that this is a preliminary study. In your response you wrote that you had changed this after my comment to the previous version of the manuscript. I don't think this study is preliminary. Maybe a better word is "exploratory".

**Response:**

Yes Exploratory is better.

After your comment I had changed

"Based on our results we can make an preliminary assessment"

To

"Based on our results we can make an assessment"

But I missed this one…thanks!

**Editor:**

P14, L29 … its output limits direct application …

**Response:**

Corrected

**Editor:**

P16, L21 Wegener (typo)

**Response:**

Thank-you, Corrected

Thanks and best wishes

Mario

**What are the prospects for seasonal prediction of the marine environment of the Northwest European shelf?**

Jonathan Tinker[1], Justin Krijnen[1], Richard Wood[1], Rosa Barciela[1], Stephen R. Dye[2,3].

[1] Met Office Hadley Centre, Exeter, EX1 3PB, UK.

[2] Cefas, Lowestoft, NR33 0HT, UK.

[3] School of Environmental Sciences, University of East Anglia, Norwich, NR4 7TJ, UK.

Correspondence to: Jonathan Tinker (jonathan.tinker@metoffice.gov.uk)

**Abstract.**

Sustainable management and utilisation of the Northwest European Shelf seas (NWS) could benefit from reliable forecasts of the marine environment on monthly-to-seasonal timescales. Recent advances in global seasonal forecast systems, and regional marine reanalyses for the NWS, allow us to investigate the potential for seasonal forecasts of the state of the NWS. We identify three possible approaches to address this issue: A) basing NWS seasonal forecasts directly on output from the Met Office's GloSea5 global seasonal forecast system; B) developing empirical downscaling relationships between large-scale climate drivers predicted by GloSea5, and the state of the NWS; and C) dynamically downscaling GloSea5 using a regional model. We show that the GloSea5 system can be inadequate for simulating the NWS directly (approach A) and so move on from A). Turning to B), wWe explore empirical relationships between the winter North Atlantic Oscillation (NAO), and NWS variables estimated using a regional reanalysis (approach B). We find some statistically significant relationships, and present a skilful prototype seasonal forecast for English Channel sea surface temperature. We then consider the potential of C). We find large-scale relationships between inter-annual variability in the boundary conditions and inter-annual variability modelled on the shelf, suggesting that dynamic downscaling may be possible (approach C). We also show that for some variables there are opposing mechanisms correlated to the NAO, for which dynamic downscaling may improve on the skill possible with empirical forecasts. We conclude that there is potential for the development of reliable seasonal forecasts for the NWS, and consider the research priorities for their development.

**Copyright Statement**

The works published in this journal are distributed under the Creative Commons Attribution 4.0 License. This licence does not affect the Crown copyright work, which is re-usable under the Open Government Licence (OGL). The Creative Commons Attribution 4.0 License and the OGL are interoperable and do not conflict with, reduce or limit each other.

© Crown copyright 2018

**1. Introduction**

**1.1. Background**

The Northwest European Shelf seas (NWS) are of wide economic, environmental and political importance. They support many ecosystem services and human activities, including fisheries, energy extraction and transmission (both renewable and non-renewable), shipping and waste removal. Most of these services and activities are sensitive to the variable environmental conditions under which they operate, for example:

- Shipping (transport and industrial) and offshore oil/gas and renewable operations are sensitive to wind/wave conditions and currents;
* * *
**Comment [j1]:** P1, L11 We show that the GloSea5 system is inadequate for simulating the NWS directly.
("can be" is strange here) (delete the last part of the sentence: and so move on from A).

**Comment [j2]:** P1, L15 I suggest: As to point C) we find large-scale …

[revised manuscript text omitted]

**Comment [j3]:** P5, L12 This is the first time SSS is used. They should be defined here.

limited observed time-series to assess its performance at replicating several observed events. Here we describe the observed time-series.

**2.4.1. Southern North Sea Ferry data**

Ferries are well established vessels of opportunity for oceanographic measurements taking regular long-term samples of
5  surface water while the ship is on passage between ports (Bean et al., 2017). Observations can be in the form of samples taken by crew for subsequent testing in a laboratory or more sophisticated "Ferry boxes" as packages of instruments that semi-autonomously monitor temperature, salinity and other water properties. We use the monthly salinity data from the ferry on the Harwich to Hook of Holland route (see 'S' in Figure 1a), which took quasi-weekly temperature and salinity samples at 9 standard stations between 1971 and 2012 (Joyce, 2006) and is reported in the ICES Report on Ocean Climate (Larsen et
10  al., 2016) and MCCIP Report Cards (Dye et al., 2013). We use point time-series from this dataset to compare to the model.

**2.4.2. Western Channel Observatory (WCO)**

The Western Channel Observatory (WCO) is an oceanographic time-series in the Western English Channel (Smyth et al., 2015). *In situ* measurements are undertaken fortnightly at open shelf station E1 (50.03°N, 4.37°W, see 'W' in Figure 1a) using the research vessels of the Plymouth Marine Laboratory and the Marine Biological Association. We compare time-
15  series of temperature and salinity from a range of observed depths to model output from the nearest grid box.

**2.4.3. ICES Report on Ocean Climate data**

We use annual-mean time-series data from three stations used in the ICES Report on Ocean Climate (González-Pola et al., 2018): Malin Head weather station; Fair Isle; and Helgoland Roads. Malin Head SST (55.37°N 7.34°W, see 'M' in Figure 1a) is provided by the Irish Marine Institute/Met Éireann (Cannaby and Hüsrevoğlu, 2009; Nolan et al., 2010). The Fair Isle
20  time-series (59°N 2°W, see 'F' in Figure 1a) is provided by Marine Scotland Science to measure the temperature and salinity (upper 100m) of Atlantic water entering the North Sea between Scotland and the Shetland Islands, via the Fair Isle Current (Hughes et al., 2018). The Helgoland Roads (54.1833°N 7.9°E, see 'H' in Figure 1a) time-series is provide by the Alfred Wegnenger Institut/Helmholtz-Zentrum für Polar und Meeresforschung, and comprises of surface temperature and salinity (Raabe and Wiltshire, 2009; Wiltshire et al., 2015; Wiltshire and Manly, 2004). The data are freely available to download
25  from ICES (https://ocean.ices.dk/iroc/).

**2.4.4. NAO**

The North Atlantic Oscillation (NAO) is a climatic phenomenon in the North Atlantic Ocean of fluctuations in the difference of atmospheric pressure at sea level between the Icelandic low and the Azores high. These fluctuations control the strength and direction of westerly winds and storm tracks across Europe (Hurrell, 1995).
30  We use the NOAA National Weather Service Climate Prediction Center NAO data (http://www.cpc.noaa.gov/products/precip/CWlink/pna/nao.shtml), for monthly mean NAO index. We only use winter (DJF) for the years 1992/1993 – 2010/2011 to be consistent with the available GloSea5 ocean and sea-ice reanalysis NAO index time-series.

**2.4.5. Storm track latitude index**

35  When analysing the relationships of the shelf salinity we find correlation patterns which suggest storm track latitude may be important. We therefore analysed the mean sea level pressure data to produce a Storm track latitude index, following a method adapted from Lowe et al. (2009). The 3-hourly ERA Interim mean sea level pressure data from all modelled latitudes at 2°30'E were filtered with a Blackman band pass filter. The temporal variance of this filtered mean sea-level pressure was

**Comment [j4]:** P6, §2.4.2 and 2.4.3 You are using data from particular locations on the NWS. Some of the time series stations are shown in Fig. 5, but other data (for example, Hoek van Holland-Harwich) are not. Please show them in one of the figures.

**Comment [j5]:** P6, L12 Wegener (typo)

[revised manuscript text omitted]

**Comment [j9]:** P14, L19-26 Many times the word "may" is used here. I think in many cases this is unnecessary and tends to give the reader the impression that nothing is certain or even doubtful. Similar for some cases in the Conclusion section.

**Comment [j10]:** P14, L28 Here it is stated that this is a preliminary study. In your response you wrote that you had changed this after my comment to the previous version of the manuscript. I don't think this study is preliminary. Maybe a better word is "exploratory".

**Comment [j11]:** P14, L29 … its output limits direct application …

[revised manuscript text omitted]